# Past and future dynamics of the Brunt Ice Shelf from seabed bathymetry and ice shelf geometry

Dominic A. Hodgson[1,2*], Tom A. Jordan[1*], Jan De Rydt[3], Peter T. Fretwell[1], Samuel A. Seddon[1,4], David Becker[5], Kelly A. Hogan[1], Andrew M. Smith[1], David G. Vaughan[1]

[1]British Antarctic Survey, High Cross, Madingley Road, Cambridge, CB3 0ET, UK
[2]Department of Geography, Durham University, Durham, DH1 3LE, UK
[3]Department of Geography and Environmental Sciences, Faculty of Engineering and Environment, Northumbria University, Newcastle upon Tyne, UK
[4]Seddon Geophysical Limited, Ipswich, UK
[5]Physical and Satellite Geodesy, Technische Universitaet Darmstadt, Franziska-Braun-Str. 7, 64287 Darmstadt, Germany
* These authors contributed equally to this work

*Correspondence to*: Dominic A. Hodgson (daho@bas.ac.uk)

**Abstract.** The recent rapid growth of rifts in the Brunt Ice Shelf appears to signal the onset of its largest calving event since records began in 1915. The aim of this study is to determine whether this calving event will lead to a new steady state where the Brunt Ice Shelf remains in contact with the bed, or an unpinning from the bed, which could pre-dispose it to accelerated flow or possible break-up. We use a range of geophysical data to reconstruct the seafloor bathymetry and ice shelf geometry, to examine past ice sheet configurations in the Brunt Basin, and to define the present-day geometry of the contact between the Brunt Ice Shelf and the bed. Results show that during past ice advances grounded ice streams likely converged in the Brunt Basin from the south and east. As the ice retreated, it was likely pinned on at least three former grounding lines marked by topographic highs, and transverse ridges on the flanks of the basin. These may have subsequently formed pinning points for developing ice shelves. The ice shelf geometry and bathymetry measurements show that the base of the Brunt Ice Shelf now only makes contact with one of these topographic highs. This contact is limited to an area of less than 1.3 to 3 $km^2$ and results in a compressive regime that helps to maintain the ice shelf. The maximum overlap between ice shelf draft and the bathymetric high is 2-25 m, and is contingent on the presence of incorporated iceberg keels, which protrude beneath the base of the ice shelf. The future of the ice shelf depends on whether the expected calving event causes full or partial loss of contact with the bed, and whether the subsequent response causes re-grounding within a predictable period, or a loss of structural integrity resulting from properties inherited at the grounding line.

## 1 Introduction

Compilations of marine geophysical data have shown that the Coats Land ice shelves bordering the Weddell Sea in East Antarctica have historically retreated towards the grounding line once detached from pinning points on the seabed (Hodgson et al., 2018).

The Brunt Ice Shelf (BIS) is the only large ice shelf that remains intact in Coats Land south of 74°S (Fig. 1). It is formed by a series of unnamed glaciers that cross a steep grounding line at a feature known as the Brunt

40 Icefalls (75°55′S 25°0′W, Fig. 2a). This zone extends for about 80 km and marks the transition between the grounded and floating ice masses. Here the glaciers and the surrounding ice sheet calve into a 10-20 km wide zone of icebergs (sometimes referred to as the ice mélange, Fig. 2a) which are eventually fused together by sea ice and falling and drifting snow layers to form a structurally heterogeneous ice shelf (King et al., 2018).

 The BIS is bounded to the north by the Stancomb-Wills Glacier Tongue (SWGT, Fig 1a), and to the south west

45 by the Dawson Lambton Glacier Tongue (DLGT, Fig 1a, 2b). Together they form a continuous floating ice mass of approximately 33,000 km$^2$ (De Rydt et al., 2018). Halley VIa Research Station is currently situated on the BIS at 75°34'S, 25°28'W, having been recently moved from the Halley VI site at 75°36′S, 26°12′W due to the development of rifts in the ice shelf. This base is the latest in a series of six British research stations that have occupied the ice shelf since 1956 (Fig. 1b). A programme of glaciological monitoring supports the operational

50 activities and, as a result, the BIS is one of the most intensely observed ice shelves in Antarctica. Since the 1956-58 International Geophysical Year, glaciological experiments, aerial photographs, and more recently satellite images, in situ radar surveys, and a network GPS stations, have all been used to monitor the behaviour of the ice shelf (De Rydt et al., 2018, Gudmundsson et al. 2016). The changing positions of the ice front have also been mapped since the Shackleton-led Endurance expedition into the Weddell Sea in 1915 (Worsley, 1921).

55 The historical records show cyclical changes of both the SWGT and the DLGT over the last 100 years. These have included an advance, then substantial calving of the SWGT sometime between 1915 and 1955 (Thomas, 1973) and the formation and loss of several >10 km-long ice tongues formed by the Dawson-Lambton Ice Stream since 1958 (Admiralty Charts, British Antarctic Survey Archives). In both cases, these glacier tongues have been of insufficient extent or thickness to re-establish contact with the topographic highs mapped at the

60 distal ends of their glacial troughs (Hodgson et al., 2018). This failure to reconnect with the bed means that they remain un-buttressed and have been described as 'failed ice shelves' (Hodgson et al., 2018) (Fig. 2b).

 In contrast, the BIS has only experienced relatively small episodic calving events along its ice front since 1915 (Anderson et al., 2014). This relative stability has been the result of the base of the ice shelf maintaining contact with a topographic high on a ridge of glacial sediments (Seismic profile 5; Elverhøi and Maisey, 1983) known

65 as the McDonald Bank (75°28′S 26°18′W). This topographic high deflects the flow of the SWGT to the north and BIS to the south. Back stresses from the contact between the base of the ice shelf and the bed buttress the ice shelf and create a series of upstream concentric pressure waves with extensive crevassing and rifting known as the McDonald Ice Rumples (MIR, Figs. and, 2c). These pressure waves rise up to c. 10 m above the surrounding ice shelf. Downstream of the MIR there is a channel of open water and sea ice and a series of 0.1-3

70 km wide ice headlands and creeks (Anderson et al., 2014). Some of these headlands show a moderate 2-4 m rise at their seaward ends. This has been attributed to 'upwarping by the buoyancy of submarine rams of ice' extending seaward of the cliff front below the waterline (Swithinbank, 1957; page 14; Thomas, 1973; page 5).

 Although in contact with the bed at the MIR, the BIS has experienced substantial changes in its velocity during the instrumental period including periods of fast flow between the late 1970s and 2000, and 2012 to present. In

75 the late 1970s, the acceleration in ice shelf velocities from 400 to > 700 m a$^{-1}$ (Simmons and Rouse, 1984), was immediately preceded by the formation of a rift upstream of the MIR in 1968 (Thomas, 1973). This event has

been attributed either to a change in the stiffness of the ice mélange area resulting from differences in the rheology between meteoric and marine ice (Khazendar et al., 2009), or to a partial loss of buttressing caused by relatively small calving events at the ice shelf front. The latter hypothesis has been tested by an ice flow model which showed that short-term loss of mechanical contact with the bed, following the local calving event in 1971, could explain both a near instantaneous twofold increase in ice velocities over a large section of the ice shelf, and a subsequent decrease once contact was re-established (Gudmundsson et al., 2016).

Recent GPS measurements of ice velocity have revealed an ongoing and spatially heterogeneous acceleration in the flow of the BIS since 2012 (Gudmundsson et al., 2016). The rapid propagation of a new rift immediately upstream of the MIR in October 2016 (Fig 1b) has led to further changes in the velocity of the ice shelf. Known as the 'Halloween Crack' this feature now extends to the northeast, partly decoupling the BIS from the SWGT (De Rydt et al., 2018). At the same time, an existing rift in the ice shelf, known as 'Chasm 1' has been widening from a few cm/day at the tip to >20 cm/day elsewhere (Fig. 1b). It is now over a kilometre wide at the southern edge of the ice shelf, and its tip is propagating north towards the MIR at varying rates of up to 4 km yr$^{-1}$, with occasional episodes of rapid propagation controlled by the heterogeneous internal structure of the ice shelf (De Rydt et al., 2018; King et al., 2018). When this reaches the MIR, or connects with the Halloween Crack it is expected that an iceberg will form and float away to the west. This will be the largest calving event on the BIS since the start of observations.

The first aim of this study was to describe past changes in ice sheet configurations in the Brunt Basin based on geomorphological interpretations of the bathymetry and subglacial topography. The second was to combine these data with surveys of ice shelf geometry to examine the present-day contact between the BIS and the McDonald Bank, to evaluate a range of future ice shelf configurations following the calving event including: (1) a new steady state of the BIS where it remains in contact with McDonald Bank; (2) a loss of contact followed by accelerated ice shelf flow and re-grounding; (3) the formation of an unpinned, but structurally viable glacier tongue resulting from (temporary, or longer term) loss of contact with the McDonald Bank; or (4) a catastrophic breakup resulting from the removal of buttressing coupled with a loss of structural integrity. These four different outcomes depend initially on the direction of propagation of Chasm 1. More precisely, whether Chasm 1 propagates to a point at or downstream of the MIR, thereby maintaining contact between the ice shelf and the bed, or joins the Halloween Crack upstream of the MIR, potentially unpinning the BIS from the McDonald Bank. Model predictions based on the present-day stress distribution of the ice shelf suggest a future pathway that leads directly to the centre of the MIR (De Rydt et al. 2018).

Critical to differentiating between these outcomes is determining the nature and geometry of the contact between seafloor and the base of the ice shelf. Here we use a range of geophysical data to build an understanding of the regional seafloor bathymetry and subglacial topography, including under the floating ice of the BIS and SWGT and their grounded ice sheet catchments. We combine this with measurements of current ice shelf geometry. This includes digital elevation models to define the area of the MIR and derive estimates of the ice shelf draft, focusing on ice shelf flow lines upstream of the MIR. These provide the basis for an analysis of the current interaction between the ice shelf and the bed, and of the future development of the ice shelf following the calving event.

## 2 Methods

### 2.1 Bathymetry and subglacial topography

In open water areas, the seabed bathymetry was derived from compilations of ship based multibeam and single beam bathymetric data (presented in Hodgson et al., 2018) combined with the International Bathymetric Chart of the Southern Ocean (IBSCO; Arndt et al., 2013).

In inaccessible areas that are covered by the ice shelf, the seabed bathymetry was derived from bathymetric measurements from historical ship tracks inland of the present ice front, which is currently at its most advanced position since 1915 (Anderson et al., 2014). Further bathymetric control was provided by 38 seismic data points acquired from the surface of the ice shelf described in Hodgson et al. (2018). These were combined with new estimates of bathymetry from gravity and magnetic data from aero geophysical surveys in 2017 (Fig. 3). Where the ice sheet is grounded, airborne radio echo depth sounding data from BEDMAP2 (Fretwell et al., 2013) and the 2017 aerogeophysical survey were used.

Inversion of gravity data reveals the sub-ice shelf bathymetry based on the large density contrast at the water-rock interface (Cochran and Bell, 2012). However, shallow geological factors such as sedimentary basins or dense intrusions can give rise to gravity anomalies with the same amplitude and wavelength as the bathymetry, making direct inversion challenging (Brisbourne et al., 2014). By integrating gravity and aeromagnetic data to constrain the sub-surface geology, we have developed a procedure to provide the most reasonable estimate of the sub-ice shelf bathymetry in otherwise un-surveyed areas. Gravity data is from a "strapdown" type sensor which provides a resolution of ~6 km and route mean squared error of ~1.8 mGal (Jordan and Becker, 2018). This was combined with regional data from previous surveys and compilations (Aleshkova et al., 2000; Forsberg et al., 2017; Jordan et al., 2017). Coincident aeromagnetic data was used to constrain the location and size of a large mafic intrusion (Jordan and Becker, 2018), which would otherwise have significantly distorted the inversion results.

Estimation of the sub-ice shelf bathymetry from the gravity data used a four-step procedure (for details see Supplementary Note 1). First, to initiate the estimation we used interpolated grids of ice surface, sub-surface topography and bathymetry from direct observations (Figs. 3 and 4a). The gravity effect of these surfaces was calculated and subtracted from the compiled free-air gravity anomaly (Fig. 4b) to give an estimated Bouguer anomaly. The second stage was to isolate the gravity signatures in the Bouguer anomaly due to bathymetry not described by direct observation. A low pass (150 km) filter isolated signatures due to crustal thickness variations. A ~50 mGal anomaly north of the location of Halley VIa Research Station (Fig. 4b) is interpreted from magnetic and gravity data to be a large mafic intrusion 80 km long, 30 km wide and ~6km thick (Jordan and Becker, 2018). The gravity anomaly from this structure significantly distorts any bathymetric inversion. A 3D gravity model of this body was therefore constructed. Both the long wavelength crustal anomaly and the gravity model of the mafic intrusion were subtracted from the Bouguer anomaly. This provides a residual gravity anomaly due only to un-modelled variations in sub-ice shelf bathymetry, and un-modelled upper crustal

geology. The third stage of the topographic estimation process converts the residual gravity anomaly to variations in bathymetry using the Bouguer slab formula. Adding the calculated bathymetric variations back to the initial bathymetric grid gives a preliminary gravity-derived estimate of bathymetry. The fourth and final stage of the estimation process blends the bathymetry from the preliminary gravity estimate and the direct

(seismic/swath/radar) observations. To do this the gravity derived topography is forced to match observational point data where it is available, and shifted to give a smooth join with these well constrained areas, providing the best possible overall bathymetric model (Fig. 3). The final bathymetric model (Fig. 4c) retains uncertainties due to un-modelled geological structures away from direct bathymetric observations, and should not be considered the result of a formal inversion; however, it reveals the broad structure of the sub-ice shelf

bathymetry and subglacial topography.

Uncertainties arising from unknown and un-modelled geology are hard to quantify, as step 4 of the estimation procedure means the model always matches the direct bathymetric observations. One estimate of the errors due to geological factors can be made by looking at the difference between the initial gravity estimation of topography (step 3) and the direct observations. This reveals a symmetrical error distribution with ~0 mean, and

a standard deviation of ~175 m, which we attributed to un-accounted geological biases, and the long wavelength of the regional gravity data. This therefore represents a worst-case estimate of the expected error far from control points. Step 4 will have in part accounted for the impact of unknown geological features, and hence reduced the overall error of the resulting estimated bathymetry. One alternative check is to compare predictions of ice shelf flotation, based on freeboard and an assumed ice shelf density, to the final estimated bathymetry

(Fig. 4c). This reveals that the final bathymetry generally predicts the grounding line well. A key discrepancy is beneath the SWGT at 75°S, where flotation is violated by 50-100 m. We therefore consider +/-100 m a reasonable minimum estimate of the error in the bathymetric estimate in this region. Northeast of the 2017 survey area, our new bathymetric estimate suggests a broad area of ice shelf should not be floating. We attribute this to a lack of high quality gravity data coverage, and/or actual observations of bathymetry which would help

constrain the impact of un-modelled underlying geological features.

In regions where both direct topographic/bathymetric observations and high resolution gravity data are available (Fig. 4a and b respectively) major topographic structures, including the deep onshore basin and the trough beneath the BIS, are resolved as significant lows in the gravity data. This supports the use of gravity data to fill the intervening areas where no direct bathymetric measurements are available. Aeromagnetic data across the

study area (Supplementary Note 2) shows a clear high frequency signal beneath the main survey area. This suggests the geological basement is close to the surface, and a major thick sedimentary basin which could distort the results is not present. In addition, no clear 1:1 correlation between the inverted bathymetry and the underlying geologic (magnetic) fabric is seen. This supports the view that the bathymetry dominates any underlying geological signal. However, the northern part of topographic ridge 3 does appear to follow a

relatively short wavelength negative magnetic anomaly, indicating geological control of this structure. The more limited data coverage in this region (Fig. 3) makes further detailed discussion of the underlying geological origin of this feature problematic.

**2.2 Ice shelf geometry**

The surface topography of the ice shelf was measured using a high resolution surface digital elevation model
(DEM) derived from Worldview stereo imagery (De Rydt et al., 2018). A total of 7 individual Worldview tiles
with a horizontal resolution of 3 m and varying timestamps (20/10/2012 to 30/03/2014) were collated using: (1)
a surface velocity field from June 2015 [De Rydt et al., 2018] to shift individual tiles to a common datum; (2)
ground control points to fix the floating vertical coordinate; and (3) tidal corrections to correct vertical offsets.
The data was subsampled onto a 30 m x 30 m grid and cross-calibrated with airborne LIDAR data from the
2017 airborne campaign (flight lines in Fig. 3) to obtain surface elevations above sea level. Although airborne
radar measurements of ice shelf draft were available, it was difficult to accurately determine ice shelf draft with
these methods. Instead, the ice shelf draft along selected flow lines was calculated based on assumptions of a
floating ice column and the density profile of the ice, to translate the freeboard into ice thickness. A common
mid-point survey (CMP) of the top 30 m constrained the mean density in the firn pack to 750 kg/m$^3$, and
nominal ice densities of 920 kg/m$^3$ were assumed below 50 m. These density estimates were applied uniformly
across the ice shelf without local corrections for incorporated icebergs.

## 3 Results

### 3.1 Bathymetry and subglacial topography

The subglacial topography and bathymetry shows that the grounded ice occupies a complex bedrock terrain
(Fig. 4c). There is a 1900 m deep trough beneath the Stancomb-Wills Glacier with subglacial catchments to the
north, northeast and south, each fed by multiple tributary valleys. In contrast, the terrain beneath the glaciers that
discharge into the BIS consists of a series of small northwest oriented troughs that are on a bedrock surface
which generally lies <200m below sea level.

A steep coastal slope marks the transition between grounded and floating ice masses (the black and white
contour in Fig. 4c is the predicted grounding line). Downstream of this grounding line, the trough originating
under the BIS is oriented south to north, whilst the trough originating beneath the SWGT is oriented east to west
(orientations indicated by white arrows, Fig 4c). Both troughs reach depths of 600-1200 m and merge into the
Brunt Basin forming a single north northwest oriented 400-800 m deep basin that extends >120 km into the
southern Weddell Sea at 74°S. The Dawson Lambton Ice Stream occupies a small glacial trough, which extends
to the west under the DLGT (Fig. 4c).

A number of topographic highs occur in the Brunt Basin. Two of these are high enough to make contact with
floating ice at the base of the SWGT around the Lyddan Ice Rise, and at the base of the BIS forming the
McDonald Ice Rumples (Fig. 1). Other topographic highs are present in the NNW oriented part of the Brunt
Basin under the SWGT, including a series of at least three transverse ridges on the flanks of the trough (marked
as inferred grounding lines 1-3 in Fig. 4c). Our gravity-derived topography predicts these ridges are in contact
with the base of the SWGT, which is not indicated by ice velocity or satellite imagery. Instead, we suggest that
these ridges fall just short of the base of the ice. Two similar transverse ridges at the present day grounding line
in the East-West oriented part of the trough beneath the SWGT (marked as inferred grounding line 4 in Fig. 4c).

We interpret these topographic highs as likely contact points, perhaps both for grounded ice retreating from its maximum extent (grounding line positions) and stabilisation points for subsequently floating ice (pinning points).

The McDonald Bank is well resolved, particularly off shore by the swath bathymetry and single beam surveys (Fig. 4d). The east face of the McDonald Bank rises steeply from the Brunt Basin (Figs. 4d and 6). The upper
surface of the bank is relatively flat but has a number of smaller scale topographic highs, reaching a minimum depth of c. 212 m below sea level. Some of these appear to be crescent-shaped (indicated in green, Fig. 4d).

**3.2 Ice shelf geometry**

The satellite images (Fig. 1) and DEM (Fig. 5) show the heterogeneous nature of the ice shelf. The main distinction is between those parts of the ice shelf supplied by higher velocity glaciers and ice streams, and those
supplied by the low velocity ice of the inland ice sheet. The former supply the ice shelf with closely packed bands of incorporated icebergs, and the latter with icebergs that are more widely spaced. These icebergs are typically oriented with their long axis at 90° to the direction of ice flow (Fig. 1b, 5) (King et al., 2018).

The draft of the BIS was examined in detail along three flow lines (northern, central and southern) upstream of the MIR (Fig. 6). The three flow lines follow the southern edge of a moderately closely packed band of
incorporated icebergs within the ice shelf (Fig. 5). All three lines show an increase in ice shelf draft from 35 to 22 km upstream of the McDonald Bank. From 22 to 0 km the draft remains relatively constant. The base of the ice shelf is highly irregular, as a result of the keels of icebergs incorporated into the ice (King et al., 2018; Thomas, 1973).

The highest point of the McDonald Bank was 212 m below sea level, measured on the southern flow line (Fig.
6). The precise ice shelf thickness at the McDonald Bank is difficult to resolve from the radar data as there was no clear expression of the bed due to interference from the complex topography of the MIR. However, the maximum draft of the iceberg keels along the three flow lines was 214 m (Northern), 214 m (Central) and 237 m (Southern) below sea level, so the maximum potential overlap between the depth of the incorporated ice shelf keels and the depth of bed along all flow lines ranged from 2 to 25 m.

The DEM analysis of the MIR shows that the area of deformation caused by contact between the base of the ice shelf and the McDonald Bank is limited to an area of less than 1.3 to 3 km$^2$ (Fig. 5). Strain rates upstream of the MIR (Figure 8 in De Rydt et al., 2018) show the compressive regime in the ice shelf resulting from grounding in 2015.

**4 Discussion**

**4.1 Past ice sheet configurations**

The bathymetry and subglacial topography provide sufficient evidence to propose a range of past ice configurations in the study area. The substantially over-deepened south-to-north oriented trough under the BIS,

and east-to west oriented trough under the SWGT are the likely product of grounded ice streams which

converged in the Brunt Basin (white arrows, Fig. 4c) and presumably discharged westwards towards the

Filchner Trough at glacial maxima. At these times, glacial depositional processes operating between the ice

streams occupying Filchner Trough and the Brunt Basin likely formed the layered glacial sediments of the

McDonald Bank interpreted from seismic surveys (Elverhøi and Maisey, 1983). Although seismic lines at

different orientations are required to fully characterise the internal architecture and processes forming this

deposit, the line drawings based on sparker data presented by Elverhøi and Maisey (1983, Profile 5, page 486)

suggest the layered glacial sediments dip westwards into the Weddell Sea. At least two of these layers (Units 1

and 2) were interpreted as being of 'glacial origin', although whether they are 'glaciomarine sediments, till

and/or glacially compacted glaciomarine sediments, cannot be determined'. These layers have subsequently

been truncated along the steep eastern slopes of the McDonald Bank, presumably by erosive ice advances along

the south-to-north oriented trough under the BIS. The relatively flat top of the McDonald Bank may be the

product of glacial planation processes by ice shelves during, and either side of peak glacial conditions.

During the development of interglacial conditions, it is reasonable to assume that the ice stream occupying the

south-to-north oriented trough under the BIS was starved of ice. We attribute this to the relatively small

northwest oriented glaciers in its catchment being largely <200m below sea level and therefore susceptible to

progressive isolation from the deep troughs of the inland ice sheet. Instead, we suggest that most of the regional

ice flow progressively followed the deep trough under the Stancomb-Wills Glacier, channelling ice from several

north-, northeast- and south-oriented subglacial catchments, before discharging it west into the Brunt Basin and

then north-northwest towards the southern Weddell Sea.

The topographic highs and ridges at, and just south of the 74°S parallel, and at the 75°S parallel (labelled 1-3 in

Fig. 4c) mark at least three potential former potential grounding line positions. These are similar to transverse

ridges and other topographic highs at the present day grounding line (labelled 4 in Fig. 4c). These inferred

grounding lines are separated by deep trough basins, so it is reasonable to assume that during deglaciations the

ice sheet stepped back through a series of grounding lines at these topographic highs.

Following grounding line retreat, floating ice likely occupied the Brunt Basin, as it does today. In this

configuration, the various topographic highs associated with the highest parts of the transverse ridges and

former grounding lines, and the McDonald Bank would have formed potential pinning points for advancing ice

shelves and glacier tongues. Most of these features fall just short of the base of the ice. Although the McDonald

Bank has a relatively flat top, contact between the ice and the bed can be inferred from evidence of smaller-scale

surface topography, including the poorly resolved crescent-shaped features (Fig. 4d). The latter may be ice-push

moraine complexes formed where dense aggregations of deep keels from icebergs incorporated into the ice shelf

from upstream glacial troughs have grounded on the bank.

As the BIS continued to thin, the topographic highs on the McDonald Bank would have provided the last

potential pinning points and frontal buttressing of advancing ice. In its present configuration, only one of these

is high enough (c. 212 m below sea level) to maintain contact with the base of the BIS at the MIR. In contrast,

the SWGT is neither thick nor extensive enough to ground on the McDonald Bank but may benefit from lateral

buttressing from the Lyddan Ice Rise to the northeast, and from the McDonald Bank during periods when it is coupled with the BIS (it is presently decoupled by the Halloween Crack (De Rydt et al., 2018)). The DLGT is also not thick enough to ground on the grounding zone wedge at the seaward end of the Dawson Lambton trough, although its internal architecture suggests grounding in the past and the presence of a frontally buttressed ice shelf (Hodgson et al., 2018).

**4.2 Contact between the Brunt Ice Shelf and the bed**

Satellite images and aerial photographs show that the MIR have changed in their extent and morphology at different times in the recent past. This can be interpreted as an indicator of periodic thinning and at least partial loss of contact with the bed resulting from changes in ice shelf draft and minor calving events (cf. Bindschadler, 2002). The BIS has significant local thickness variations. This means that along any flowline the ice shelf draft varies considerably. This is due to a combination of the presence of incorporated iceberg keels, initial ice thickness close to the grounding line, changes in mass balance from accumulation of snow on the surface, basal accretion of marine ice, compression along the flow line from the McDonald Bank, ice velocity and local flow-divergence. Below, we consider how these influence the draft of ice shelf approaching the MIR.

Changes in mass balance from snow accumulation, and accretion of marine ice can be estimated from meteorological records and published over-snow radar surveys of the internal structure of the ice shelf (King et al., 2018). Meteorological records show a mean snow accumulation rate (Halley Station data from 1972-2017) of 90 cm/yr (range 48-149 cm). King et al. (2018) have shown that the addition of this snow and firn accumulation drives the incorporated meteoric icebergs below sea level and explains the increase in the ice shelf draft away from the grounding line (Fig. 6). From this point (22 km upstream of the MIR) there is no further downstream increase in ice shelf thickness that can be attributed to firn accumulation or the accretion of marine ice or other processes such as compression (Fig. 6). This may be due to lateral spreading under gravity, temporal changes in the stress regime as the ice shelf goes through phases of well buttressed and lightly buttressed conditions, or basal melting. The latter is likely to be of limited magnitude (< 0.5 m/yr) based on measurements of seawater temperatures under the Riiser-Larsen Ice Shelf immediately to the north of the Lyddan Ice Rise (Gjessing and Wold, 1979). However, the sub-ice shelf bathymetry presents no substantial barriers to the future penetration of warm circumpolar deep water that has been predicted, by one model, in the second half of the 21st Century (Hellmer et al., 2012).

The ice shelf flow line analysis shows no evidence of upstream thickening of the ice shelf resulting from buttressing (Fig. 6), although the modelling suggests that the compressive regime is felt at least 70 km upstream of the MIR (De Rydt et al., 2018). Analysis of the strain rate patterns using ice velocities, shows that outside of the local area of compression, most of the ice shelf is moving in similar directions at similar velocities with thinning rates of less than 1m/yr as a result of flow divergence (Figure 8 in De Rydt et al., 2018). However, this thinning signal is largely offset by surface accumulation. Thus, from at least 22 km upstream of the MIR there are no overall trends in thickness resulting from natural ice-shelf evolution down the flow line (Fig. 6).

The flow line analysis also shows that the base of the ice shelf is highly heterogeneous as a result of the incorporated iceberg keels. Maximum keel depths along our flow lines ranged from 214 m (Northern and

Central) to 237 m (Southern). This supports ice-penetrating radar analysis of the internal architecture elsewhere on the ice shelf that shows keels ranging between 175-250 m depth interspersed with thinner sections of ice

shelf formed by accumulated sea ice and sea-water-saturated firn, snow fall and drift (Fig. 4 in King et al., 2018). At least some of the cracks and chasms downstream of the MIR may be the expression of the differential strain experienced during the grounding, then release of the incorporated icebergs, versus the intervening brine and firn sections of ice shelf (this is most apparent downstream of the MIR on the northern side (See Fig. 2 in King et al., 2018)).

Collectively the ice shelf geometry analyses suggest that the distribution and depth of the incorporated iceberg keels is key to determining the future grounding potential of the ice shelf it flows towards, and around the MIR. Their distribution can be inferred from the surface topography and ICESat data (Fig. 5), as well as Sentinel-1 satellite radar images (King et al., 2018). This shows areas on flow lines upstream of the MIR where the icebergs are more widely spaced or have shallower keels that could fail to make contact with the McDonald

Bank (Fig. 6).

**4.3 Future evolution of the Brunt Ice Shelf**

The future integrity of the BIS depends not only on the physical properties of the ice, which are poorly understood, but also on whether it remains grounded on the McDonald Bank. In the immediate future, this will be determined by the direction of propagation of the tip of Chasm 1, the dynamics and geometry of the expected

calving event and the subsequent response of the remaining ice shelf. We consider four scenarios below:

1. Chasm 1 propagates to, or downstream of, the MIR and the BIS remains pinned to the McDonald Bank following the calving event. In this scenario, a new steady state will persist as long as the ice draft and keels are deep enough to maintain contact with the bed. This situation has persisted since the establishment of Halley

Research Station on the ice shelf in 1956, and has included the short-term loss and reestablishment of mechanical contact with the bed, following a small local calving event in 1971, However, the DEM surface topography (and ICESat data) shows large variations in the distribution of the incorporated icebergs (Fig. 5), and the ice shelf draft upstream of the MIR (Fig. 6) which can be used to determine if grounding will be maintained in future. For example, there are currently portions of the ice shelf, due to pass over MIR 4-12 years

from 2017 that may have insufficient draft to ground, with no keels extending below -200 m. At this time the ice shelf might be more prone to reduced, or loss of, contact (Fig. 6b).

2. Chasm 1 propagates to, or upstream of, the MIR, the BIS temporarily loses contact with the bed, but then rapidly increases in velocity and re stabilises. If Chasm 1 joins the Halloween Crack at or upstream of the McDonald Bank then contact with the bed will be reduced, or lost following the calving event. In this scenario

an immediate increase in the velocity in the ice shelf might be anticipated as a result of the removal of buttressing (Gudmundsson et al., 2016). This 'surge' could result in a re-grounding on the McDonald Bank and re-stabilisation of the ice shelf.

3. Chasm 1 propagates upstream of the MIR, the BIS loses contact with the bed, and this ungrounded state persists for sufficient period that the ice shelf essentially becomes a unpinned glacier tongue, which may or may not extend beyond the McDonald Bank, but does not reground. The analogue for this scenario is the SWGT, which has maintained a degree of structural integrity in the absence of frontal buttressing (Fig. 1a), but experiences large calving events at the ice front. The SWGT extends more than 200 km from the grounding line, if BIS were to follow this scenario it could advance forward, and potentially reground on topographic highs elsewhere on the McDonald Bank (including at the location of the MIR), but the timescale for a re-grounding remains highly uncertain.

Under scenarios 1-3 the presence of the large iceberg calved from the BIS, poses an additional threat to the integrity of the remaining parts of the ice shelf through iceberg collision. This hypothesis was considered by Anderson et al. (2014) with respect to a calving event on the SWGT.

4. Catastrophic breakup resulting from the removal of buttressing coupled with a loss of structural integrity. In this scenario, Chasm 1 propagates upstream of the MIR, the BIS loses contact with the bed, and the subsequent increase in velocity is not extensive enough for the ice shelf to reconnect with the bed, but is sufficient to increase strain rates to a level where further fractures across the ice shelf are likely. This more widespread collapse would likely be influenced by the internal structural properties inherited at the grounding line (King et al., 2018). Observations show that these have resulted in occasional episodes of fast crack propagation (De Rydt et al., 2018), and inverse and forward modelling results suggest vulnerability to destabilisation by relatively rapid changes in the ice mélange properties, resulting from the interaction of its marine ice component with ocean water, or by the further propagation of a frontal rift (Khazendar et al., 2009). The local analogues for a more widespread collapse scenario are the DLGT, which is a glacier tongue that is subject to frequent calving events (Fig. 2c), and the ice margin south of the BIS that calves directly into the Weddell Sea (Fig. 2d).

Although the outcome of the calving of the ice shelf is not yet known, these four potential scenarios all show the importance of understanding the geometry the ice shelf and the bed. Which of the four scenarios will transpire depends, at least in part, on whether the ice shelf is able to remain in contact with one of the topographic highs on the McDonald Bank. Whilst the depth of the topographic highs is fixed, the contact between the ice shelf and the bed depends on a number of variables. These include the short-term development of the ice shelf flow regime following calving, and its impact on ice shelf draft (e.g. lateral spreading) and structural integrity (e.g. development of further rifts resulting from changes in the strain rate). They also include the long-term influences of processes at the grounding line, such as the thickness and velocity of ice flowing across the Brunt Icefalls, which determine the spacing and depth of the iceberg keels.

**5 Conclusions**

The ice dynamics of the BIS and SWGT glacier catchments have evolved from widespread occupation by grounded ice streams (during glacial periods), to a retreat of grounded ice through several inferred grounding line positions (during deglaciation), followed by the development of floating ice forming both buttressed ice shelves (BIS) and glacier tongues (SWGT, DLGT). In the former case, buttressing has occurred where the ice has been thick enough to maintain contact with topographic highs in the seabed.

The BIS has maintained its overall structural integrity since it was first observed in 1915, despite experiencing several periods of fast flow (Simmons and Rouse, 1984), fracturing, and episodic calving events along its ice front (Anderson et al., 2014). This relative stability can be attributed to its (at least intermittent) grounding on the McDonald Bank.

Although situated in a region of Antarctica that is not presently experiencing a rapid increase in atmospheric
temperature or a known intrusion of circumpolar deep water, the BIS has nonetheless entered a period of rapid change (De Rydt et al., 2018), marked by the rapid propagation of rifts that will likely result in the largest calving event since observations began. Being composed of icebergs fused together by sea ice and accumulated snow, its internal structure differs from most other Antarctic ice shelves, and hence its dynamics are more difficult to predict. Specifically, it is not yet known if the ice shelf will lose contact with the bed and how it will
respond after the calving event. Based on the history of different ice sheet configurations and the geometry of the ice shelf and the seabed, we have outlined four scenarios that might occur following the expected calving event, which will occur as Chasm 1 progresses north. These scenarios range from a re-stabilisation of the BIS to a more widespread collapse.

Priorities for future work on the BIS include: (1) Continuing to assess changes in the flow velocity and
compressive regime in the ice shelf resulting from ungrounding at the MIR. (2) Using radar data to examine changes in ice shelf draft resulting from compression and changes in mass balance (firn and snow accumulation and marine ice accretion); specifically, modelling the balance between accumulation and lateral spreading under different grounding scenarios. (3) Calibration and refinement of the geometry data by direct measurements of seafloor bathymetry in new areas exposed by iceberg calving, and measurements of ice shelf thickness and
depth to the bed where the BIS meets the McDonald Bank (with access through sea ice in the Halloween Crack and Chasm 1). (4) Further measurements to assess the different material properties of incorporated-icebergs versus the intervening areas of sea ice and accumulated snow (temperature, viscosity, fracture toughness etc.) (5) An assessment of the future evolution of the ice thickness and draft from transient model runs with assumptions about the present and future calving events.

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

**Figures**

**Figure 1. Study location map over Landsat-8 satellite images. (a) Composite of visible band Landsat-8 satellite images showing the Brunt Ice Shelf and Stancomb-Wills Glacier Tongue. Pale blue shading highlights the current cracks and Chasms seen on the ice shelf. Camera symbols approximately locate photographs in Fig. 2a-c. Red box locates panel (b) and inset shows the study area (red) in its Antarctic context. (b) Detail of the Brunt Ice Shelf showing the Halloween Crack and Chasm 1. Landsat-8 images from the 5th, 9th, and 12th of February 2018 courtesy of the U.S. Geological Survey.**

**Figure 2. Images of key ice shelf features. Approximate locations of a-c in Fig. 1. (a) The grounding line of the Brunt Ice Shelf showing the 10-20 km wide zone of icebergs which eventually 'fuse' together with sea ice and falling and drifting snow layers to form a structurally heterogeneous ice shelf. (b) The Dawson Lambton Glacier Tongue, which fails to reconnect with the bed. (c) The McDonald Ice Rumples showing where the ice shelf is deflected around a topographic high on the McDonald Bank, oblique aerial photograph facing NNW in 2017. (d) Part of the Coats Land ice margin south of the Brunt Ice Shelf, which calves directly into the Weddell Sea. Photograph taken of the coast immediately south of the study area in Fig. 1a.**

**Figure 3. Data locations overlain on a MOA satellite image. Yellow lines indicate 2017 aerogeophysical flights collecting radar, gravity and magnetic data. Red shading indicates swath bathymetry data coverage. Blue lines mark radar depth determinations from BEDMAP2 (Fretwell et al., 2013). Orange lines mark location of ICEGRAV 2013 radar and gravity data (Forsberg et al., 2017). Black dots mark seismic determinations of water column thickness under the Brunt ice Shelf (Hodgson et al., 2018). White dots mark soundings from historical ship tracks acquired when the ice front of the Stancomb-Wills Glacier Tongue was less-advanced. White line marks the grounding line/ice shelf edge from the 2013 Antarctic Digital Database. Red square marks approximate location of Halley VIa Station.**

**Figure 4. Revised topography beneath the Brunt Ice Shelf and Stancomb-Wills Glacier Tongue. (a) Topography derived from direct observations including; swath bathymetry offshore and in areas historically accessible to ships during past calving events, seismic depth sounding of the ice shelf, and radar depth sounding over the grounded ice sheet. (b) Free air gravity anomalies. Data inside black outline from new strapdown gravity data set (Becker et al., 2018). Regional data from ICEGRAV 2013 survey (Forsberg et al., 2017) and previous regional compilations (Jordan et al., 2017). Note gravity 'high' outlined in yellow attributed to a large mafic intrusion based on gravity and magnetic signatures (Jordan and Becker, 2018). (c) Integrated bathymetric model including additional constraints from gravity data beneath the ice shelf. Black and white contour is the predicted grounding line. Red contours show areas predicted to be grounded by ice 50, 100 or 200 m deeper than the calculated bed. Green contours show 50, 100, 200m predicted cavity thickness. Probable orientations of past grounded ice flow are indicated by white arrows. Inferred former grounding lines based on mapped topographic highs are marked by numbered black dashed lines (1-3) together with the current grounding line (4). (d) Detail of the**

topography beneath the Brunt ice shelf. Red lines mark the position of flow-lines plotted in Fig 6. Black dots show seismic stations and thin black lines mark edges of swath bathymetric data coverage. Green

circles and crescent-shaped lines indicate possible past and present pinning points. Abbreviations: LIR (Lyddan Ice Rise), SWS (Southern Weddell Sea), MB (McDonald Bank), SWGT (Stancomb-Wills Glacier Tongue), BIS (Brunt Ice Shelf), BB (Brunt Basin), DLGT (Dawson-Lambton Glacier Tongue).

Figure 5. High resolution surface digital elevation model (DEM) from Worldview surface height above the geoid (subsampled at 250m x 250m) for December 2016. This is cross-calibrated with airborne

LIDAR data from the 2017 airborne campaign referenced to WGS84, then corrected to the same reference grid as the DEM by applying the BEDMAP2 geoid correction to obtain surface elevations above sea level. The effective timestamp of the surface DEM is 15 January 2017. Survey flight lines are marked by thin black lines. The velocity field is represented by blue arrows, based on data from June 2015. Purple lines mark the position of the flow-lines plotted in Fig 6. White line marks the grounding line/ice

shelf edge from the 2013 Antarctic Digital Database. Background image is a Landsat 8 image from 28 January 2017. Inset shows detail of surface topography over MIR region. Colour shaded data marks full Lidar swath gridded at 10m. Note complex pattern of ridges upstream and fractures downstream from the peak elevations which mark the MIR pinning point.

Figure 6. Cross section of the McDonald Bank and the Brunt Ice Shelf showing ice shelf draft and

bathymetry along three flow lines upstream of the MIR (the position of these flow lines is marked on Fig. 4d). The flow lines are plotted both as a function of (a) distance and (b) time from the McDonald Bank calculated from feature tracking between Sentinel-1 Satellite images on 18 and 30 June 2015 (De Rydt et al., 2018). The horizontal line approximates the highest known point of the McDonald Bank. Where the ice shelf draft falls below the horizontal line we infer contact between that the ice shelf and the bed, where

it falls above the line the ice shelf will potentially become detached. This figure shows the significance of the iceberg keels in maintaining contact with the McDonald Bank. Note: the rapid increase in ice shelf draft in the 'central' flow line approaching 0 km is considered a data anomaly, which may have resulted from the complex surface topography at the McDonald Ice Rumples.

**Author contributions.** All authors contributed to the writing of the manuscript. Bathymetry data were acquired and processed by DH, TJ, PF, KH, AS, SS and DB. Data on ice shelf geometry were acquired and processed by JDR, TJ and PF.

**Data availability.** The datasets used in this paper are available at the NERC Polar Data Centre
(http://www.bas.ac.uk/data/uk-pdc/)

**Competing interests.** The authors have no conflict of interest.

**Acknowledgements.** We thank the British Antarctic Survey (BAS) Air Operations Team and airborne survey specialists including Hugh Corr, Carl Robinson and Ian Potten. Hilmar Gudmundsson (BAS and Northumbria University) for management of data acquisition on the Brunt Ice Shelf, and Ed King (BAS) for discussions on its internal structure. Steve Colwell (BAS) provided snow accumulation and temperature data. Laura Gerrish (BAS) prepared Figure 1. This research contributes to NERC grant NE/K003674/1 "Reducing the uncertainty in estimates of the sea level contribution from the westernmost part of the East Antarctic Ice Sheet". Sarah Greenwood, an anonymous reviewer and Editor Joe MacGregor are thanked for their constructive advice and suggestions.

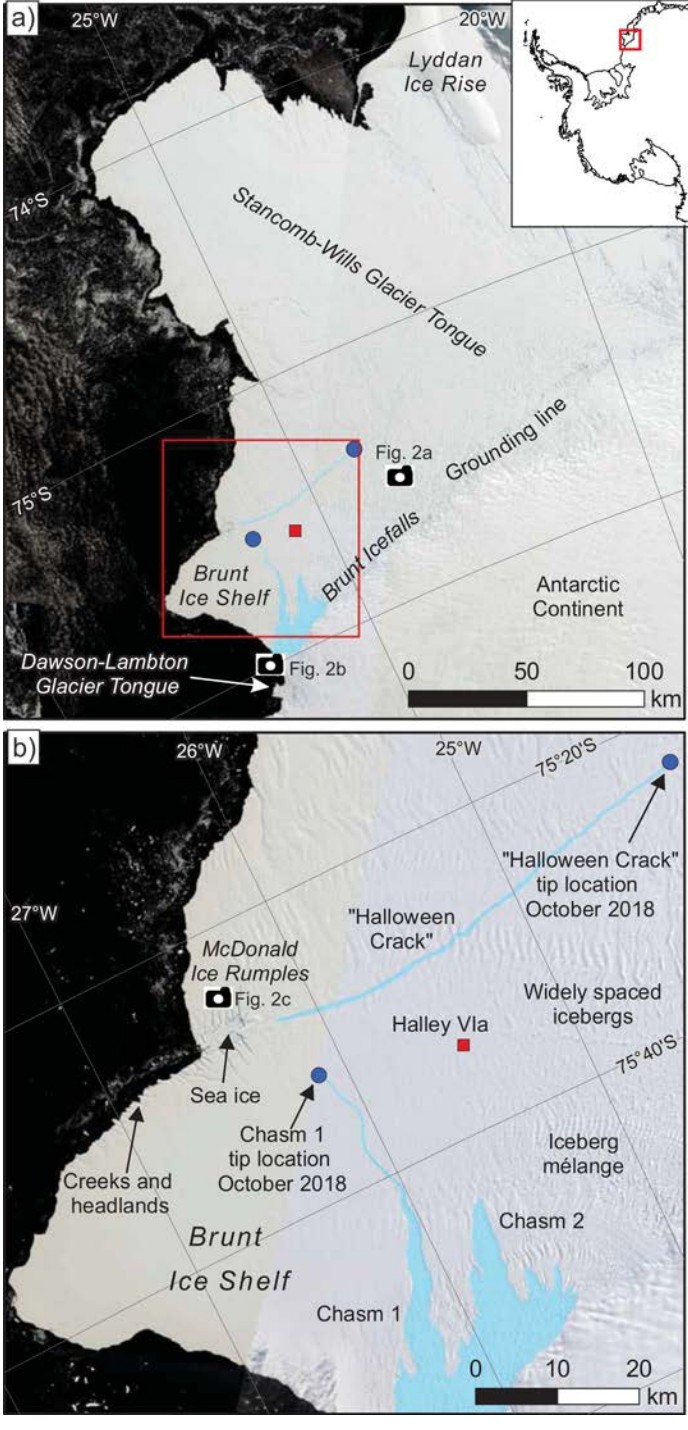

Fig.1

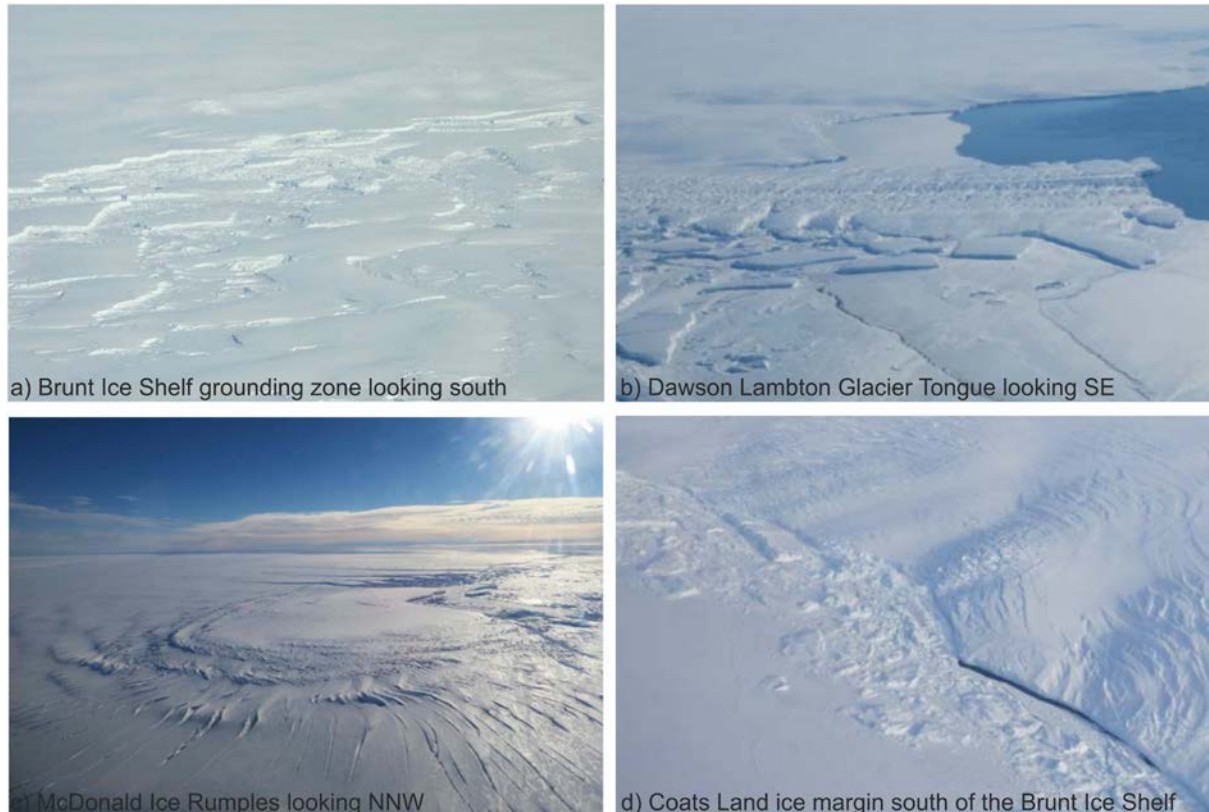


Fig. 2

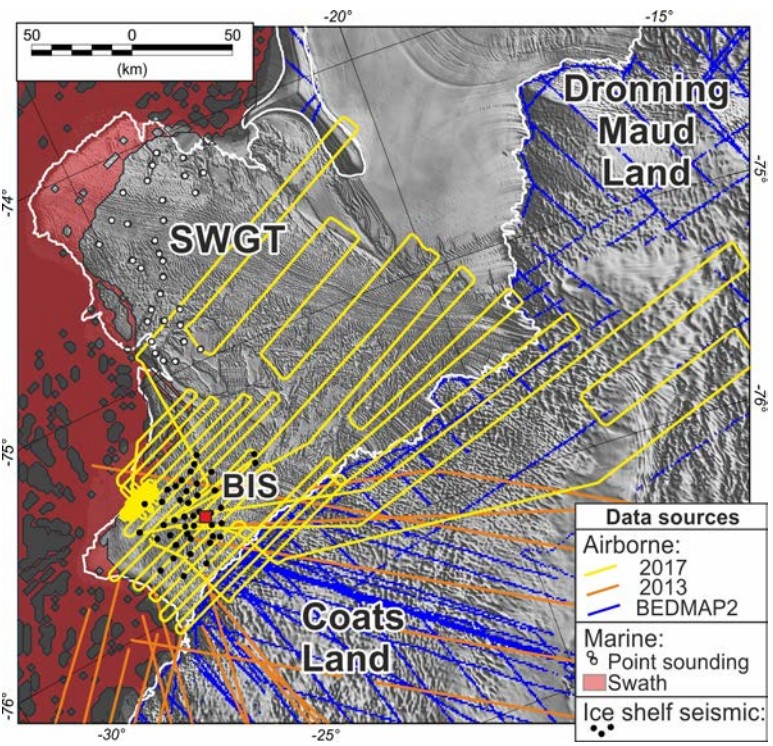

Fig. 3

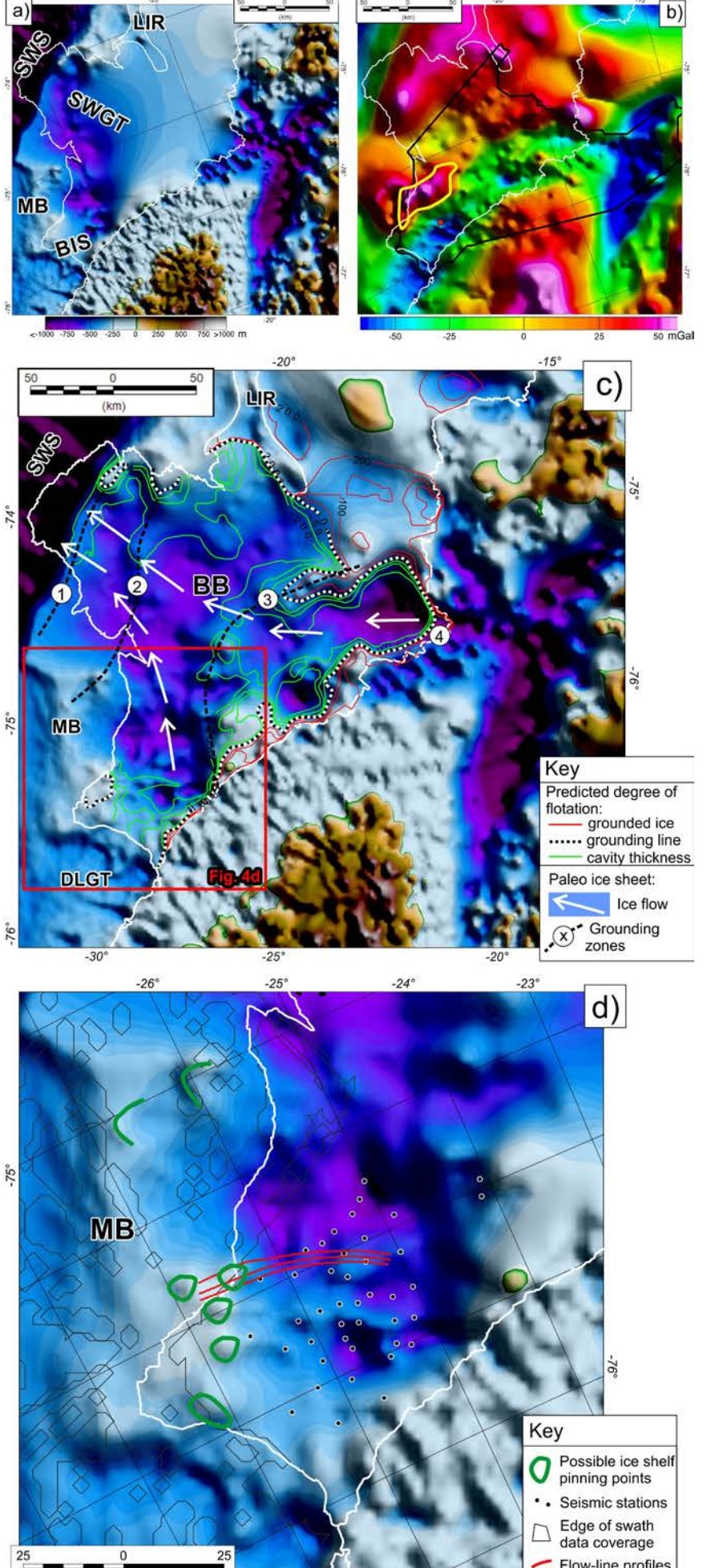

Fig. 4

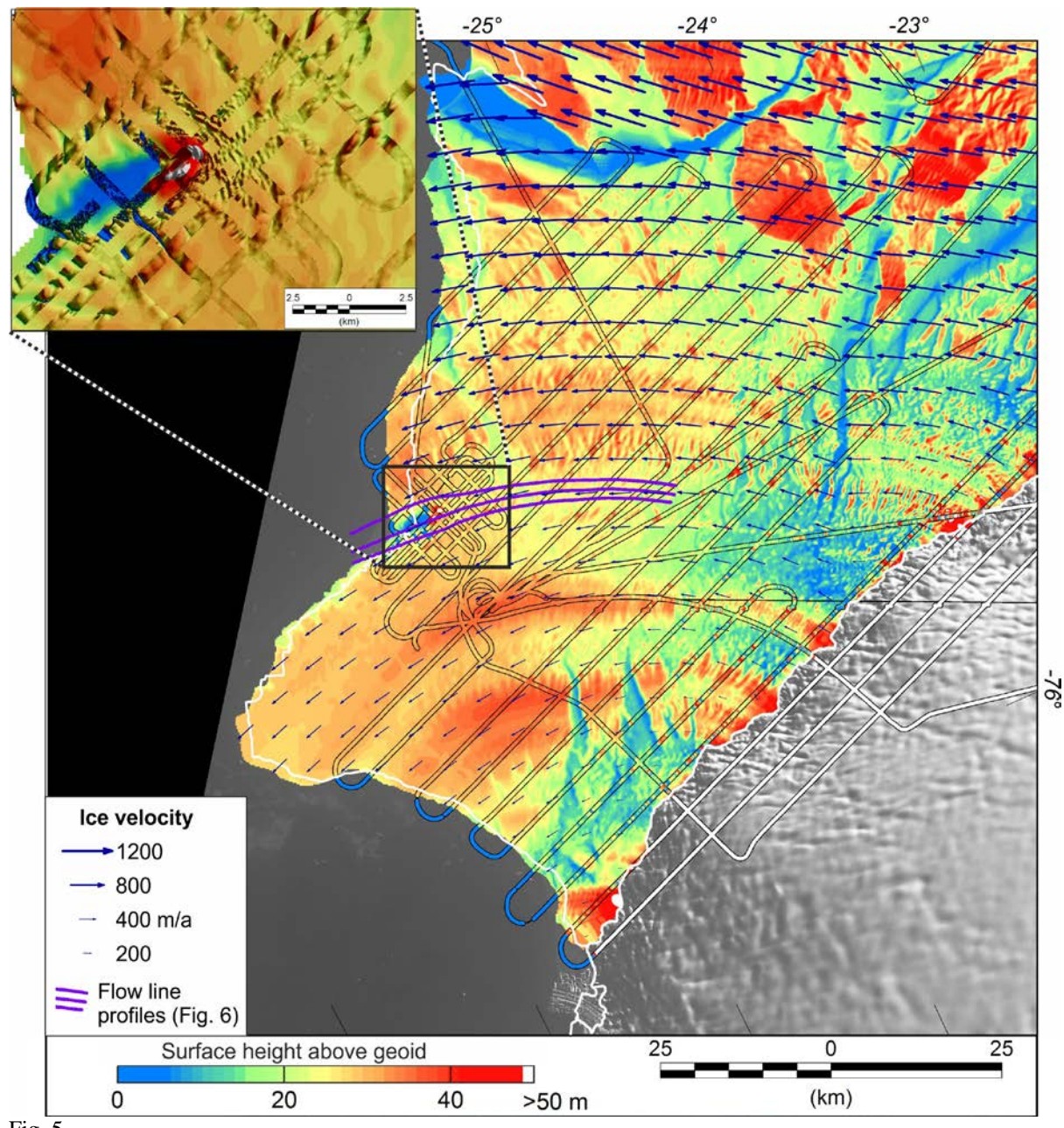

Fig. 5


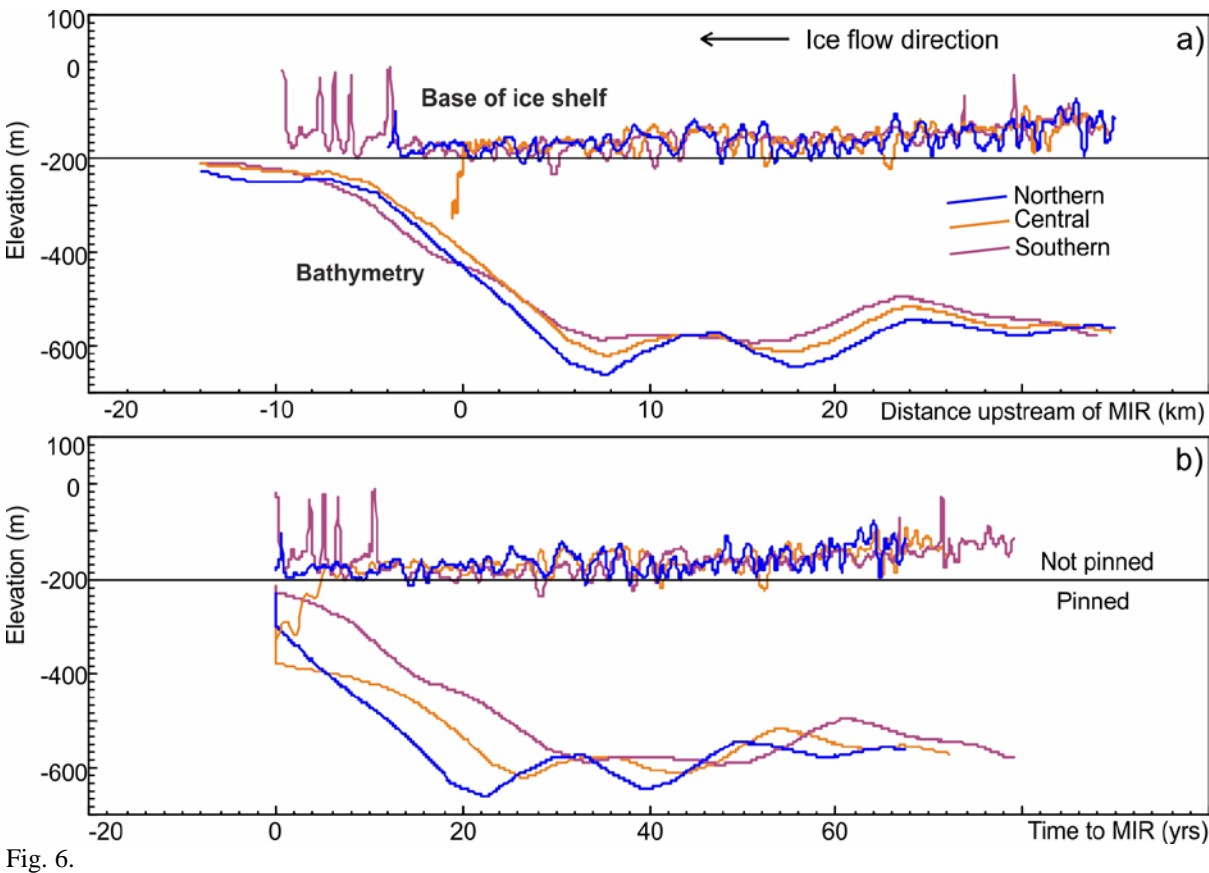

Fig. 6.