# Peer review of "Past and future dynamics of the Brunt Ice Shelf from seabed bathymetry and ice shelf geometry"

_The Cryosphere, 2018_

## Referee Comment (RC1) · Greenwood (Referee) · 26 Nov 2018

Recently developing rifts in the Brunt Ice Shelf present an opportunity to assess ice shelf responses to large calving events, and their future stability. However, in the absence of robust knowledge of sub-ice shelf topography, efforts to predict its future stability and development are limited. Here the authors integrate a range of historical, recent, proxy and direct observational data to produce the best available topographic model of the continental shelf (including sub-ice shelf). This allows them to make some crude inferences about long term glacial history in the Brunt and Stancomb-Wills catchments and, together with the topography of the ice shelf base (i.e. its along flow variability in draft) permits an assessment of the likelihood of ungrounding/regrounding on sub-ice shelf topographic highs. I found it an interesting read that raises a useful

longer-term perspective on ice sheet/shelf behaviour in the catchment, as well as insight into the local topographic/substrate relationship with shelf stability and potential feedbacks with grounding line behaviour.

However, I suggest three general elements need some attention. i) A few aspects of the methods need clarification. ii) The conclusions regarding former grounded ice flow and retreat are at best weakly supported. The interpretations are certainly plausible, even likely if we draw on other continental shelf glacial retreat systems as analogues, but here there are as yet no data to support specific reconstructions of flow or retreat. Data currently reveal the overall trough shape (coarse resolution) and modern-day ice configuration (+ recorded changes over the past ∼100 years) – we don't yet have specific glacial geological information with which to constrain the history of glacial change here. I would suggest the authors present their interpretations more as hypotheses for future testing, rather than in the tone of 'we find that the ice sheet retreated in [. . .] way'. iii) On the other hand, I thought the final aspect of the Discussion – the potential future evolution of the system – was missing an element of evaluation, and in that regard, the paper does not meet one of its specified aims. I expand on these points below, followed by some minor corrections relating to figure clarity, captions, text edits and typos.

*Methods* – a few missing aspects or in need of clarification or justification

- Stage 4 of the bathymetry reconstruction suggests the model is constrained by observational data; how do these data differ from what was used as input in Stage 1? What is left to tighten the inversion?

- Can you clarify how the ice shelf base/draft was determined? Line 106 and line 210 both suggest radar data were used; line 166 (Methods) suggests only that shelf thickness (draft) was determined using an assumed density profile and the freeboard height.

- Are density assumptions valid, since the shelf is partly composed of (fused by) marine ice?

- Explain how Fig 5 shows that contact with the MIR is limited to <1.3-3 km2 (line 215).

- Fig 5 appears to show anomalies in surface elevation between different datasets (lidar flight lines vs stereo-image DEM), particularly closer to the grounding line – can you comment on or compare the (un)reliability of these datasets? If the ice surface heights are data-dependent, what does this mean for draft calculations?

- Which velocity measurements or model were used to convert distance to time in Fig 6? Could simply be stated in the figure caption.

*Former grounded ice reconstruction*

It would be useful to see the seismic profile across the McDonald Bank (line 61, line 225 – is it possible/permitted to reproduce this figure?) and also where, specifically, it crosses the bank (e.g. to inform your comment on line 228-229). What direction do these bedded glacial sediments dip in? What grounded ice geometry would be consistent with the seismic structure – your interpreted grounded ice flow trajectory (Fig 4) would suggest the bank was lateral to ice flow reaching the continental shelf edge, but is now perpendicular to flow from the shelf (switch from lateral to ice frontal position). I wonder whether there are sufficient seismic data (coverage or resolution?) to detect this switch, or to evaluate your reconstruction? Are there any structures related to the re-grounding of the Brunt Ice Shelf on this bank? Can different generations of sedimentation be detected?

Line 185-90: I suggest removing the references to grounding line positions from the series of brackets here, and making your interpretations separately. i.e. report in these few sentences where the topographic highs are, and then finish the paragraph stating that you interpret these high points as likely pinning points, perhaps both for grounded ice retreating from its maximum extent (grounding line positions) and stabilisation points for subsequently floating ice. On first reading it seemed as if you were calling on independent evidence of grounding line positions that you now match up with your new bathymetry, but I don't think that's the case. . .

Section 4.1: since the only evidence of glacial advance/retreat that could be considered diagnostic in this new dataset are the convergent troughs, the tone of this section is somewhat over-confident and some of the assertions cannot (yet) be supported. These assertions may well be likely but specific landform or sedimentological evidence is still absent, so I would rephrase the language more towards presenting likely hypotheses than definitive conclusions.

e.g. the maximum extent of ice during the last advance is assumed to be the trough end (continental shelf break) but there is no evidence for or against this.

e.g. line 230-1: 'development of interglacial conditions' and 'become starved of ice' reads as if we can see a progression of events in the dataset (also line 247: 'continued to thin'). The progression of events is assumed, based on an assumed start point (e.g. maximum extent/flow direction based on trough size) and today's state. It might be reasonable, but I suggest making the argument based on clear observations and stated assumptions: given that the supply catchments today are much smaller than the trough size would suggest, it must be a reasonable interpretation that at some stage the extended Brunt ice stream became supply starved. (In contrast to the neighbouring Stancomb-Wills... I also note that the flow direction distal to the Brunt grounding line has gone through a ~90 degree switch, to westward – due to the much greater supply through SWG and lack of deflecting ice/topography that earlier sent discharge northward? Also worth commenting, perhaps.)

e.g. 236-240: topo highs present themselves as likely pinning points (comment on water depth, how viable would they be?) for grounding lines, and it is 'reasonable to assume' that these highs held the grounding line for a relatively longer time than the deeper basins, but we have no insight at all into whether those grounding lines might have been stable for 'significant periods'.

e.g. 241-242: how is it known that ice shelves occupied the Brunt Basin and could have stabilised on the same topo highs? This is entirely supposition. Interpretation of

push features on McDonald Bank sounds plausible, but this bank is much shallower than the other highs referred to.

Conclusion line 350: advance of shelf ice after retreat following glacial maximum is not supported; the presence of shelves/ice tongues could equally well, based on the data presented here, be a product of grounding line retreat with persistent floating ice that has gone through relatively minor cycles of calving.

The *Discussion* does not evaluate the future development of the Brunt Ice Shelf, as the article introduction states as its aim (line 94), only outlines four possibilities. A paragraph or so of synthesis seems to be lacking here, and I think there is some room for evaluation without going too far out on a limb. Whichever of the four scenarios will develop depends, at least in part, on feedbacks between ice shelf grounding on a distal pinning point, the ice shelf flow regime, the production of ice shelf (berg) keels, the draft and the packing of those keels (inherited from the shelf-sheet grounding line?) Such feedbacks are alluded to through the manuscript, but are not drawn together to inform some of the possibilities that the Discussion scenarios raise.

e.g. 199-202: velocity feedback with iceberg packing (faster supply -> more closely packed icebergs) – does this have any effect on significant keel depth, e.g. as the shelf enters a more compressive zone?

e.g. 295-6: distribution and depth of iceberg keels determines the future grounding potential of the ice shelf. What determines the distribution and depth of iceberg keels, and how would that process respond to pinning point ungrounding? Icebergs calved from a deeper grounding line should reinforce the ice shelf better, once propelled forward to the McDonald Bank. And more icebergs (more densely packed) calved from the grounding line would stabilise the shelf better. Will shelf stability feed-back with sheet-shelf grounding line position via the style/magnitude of 'calving' off the icefalls?

There is precedent for 'recovery' following a more restricted shelf (after the 70s calving event) and reduced coupling with the pinning point – but is this only in extent, or in

strength of grounding also (area of grounding or (vertical) proximity to flotation)? Based on the timeline in Fig 6b, presumably the modern ice shelf base is inherited in some way from any grounding line response to that earlier event? Can we learn anything from that?

Line 325-9: the SWG is fed by a much larger (and faster flowing?) supply basin, so presumably sustains a significant ice tongue partly due to high input, even though it's unbuttressed. Does the very different supply catchment for the BIS not make this a poor analogue?

How limited do you think your conclusions or interpretations of future ice shelf development are by the specific three flowlines you have chosen to analyse shelf draft? This is the broad flowline reaching the shallowest part of the McDonald Bank; is it also the deepest draft zone of the shelf? Could keels along other flowlines eventually reground on other topo highs on the McDonald Bank?

*Figures, annotation & captions.*

1. I wonder about the choice of satellite imagery – these show the geography clearly, but don't visualise the structures as well as, for example, Fig 1 in King et al 2018.

3: several missing features either in annotation or caption

- I don't see any shading corresponding to swath bathymetry or other sonar data (pale blue shading, in the caption).

- the white lines – calving line and grounding line (should be stated in caption) – from which source and year?

- what are the grey blobs offshore? Iceberg outlines? From what source/year, and are they really needed (could mask out)?

- could you separate the seismic sites for water depth estimates from the historical depth soundings (use different symbols)?

- what's the pink triangle? Halley VIa?

4: missing or unclear features in annotation or caption

- the depth scale isn't particularly straightforward to match to the shading on the panels, which appears more intense and has hillshade effects. Try a classified colour scale, rather than ramped, perhaps? Or make it clearer at what depth the colour bar saturates and/or at what depth the shelf break is, for reference. Also, consider shifting the blue-brown shift to 0m, rather than the unintuitive brown = submarine as well as terrestrial.

- add a label for 4d to the box in 4c

- note the black dots = seismic soundings

- remove the iceberg(?) polygon outlines, these don't seem to be necessary here

- what are the green outlined features?

- label Fig 6 flowlines on the panel itself (rather than in caption)?

- flow arrows and grounding lines are both hypothesised. Suggest something like 'Probable orientations of fast grounded ice flow. . .' and 'Inferred former grounding line positions based on existence of topographic highs. . .'

4d/5: use either decimal degrees or degrees & minutes (latitude labels) consistently on all figures.

5: label Chasm 1, 2, Halloween Crack to better follow discussion.

6: suggest label/arrow ice flow direction.

Line 191: how does Fig 4d support this statement? Info on sonar (single beam or multibeam) coverage seems to be missing from Figs 3 and 4

Line 196: are these 'smaller scale topographic highs', 'some . . . crescent-shaped' what are indicated in green on Fig 4b?

[Figure]

*Minor text edits and typos*:

Abstract language is in places unclear and in places pre-supposes some specific knowledge of the Brunt system. e.g.:

- line 16: suggest 'remains in contact with its topographic pinning points'

- line 22: suggest 'then retreated, the grounding line pausing at least three times on topographic highs and transverse flanks of the basin.'

- line 25: I find the phrase 'overlap between ice shelf thickness and the bathymetry' awkward. Having read the paper it becomes obvious what you mean, but it wasn't clear at first reading. Suggest maybe 'overlap between ice shelf draft and bathymetric highs'?

- line 26: suggest delete 'incorporated iceberg'

Line 201: later -> latter

Line 205: I assume here you mean that as we move from the 35km position to the 22km position, the shelf draft increases? I suggest writing out the 'from' and 'to' to make the direction of change clearly correspond to the rest of the statement.

Line 260: citation for the relief of 'most ice shelves'?

Line 271: 'no further downstream increase'

Line 309: what do you mean by 'occupation'?

Line 348: has -> have

---

## Referee Comment (RC2) · Anonymous Referee #2 · 27 Nov 2018

This work articulates the different possible scenarios of ice shelf stability that could follow the expected large calving event on the Brunt Ice Shelf. Authors consider the seafloor morphology of the region to infer likely configurations of ice streams in this area in the past and use this to qualitatively assess the possible future responses to calving.

The authors present a useful history of the calving, grounding and accelerations recorded during the observational period and use modeled bathymetry to infer earlier glacial configurations and predict former pinning points for the grounding line and past ice shelves. The connection between past configurations and possible future behavior is not strongly drawn, but several future scenarios are clearly articulated. The work does not distinguish between likelihoods of future scenarios but offers priorities

for future work required to investigate this question.

Scientific questions: Line 61: I assume this bank is hard-rock cored. Was the sediment thickness from seismic surveys used in the gravity bathymetry model?

Line 133 to 157: The gravity inversion technique presented here is novel, and while it makes assumptions that will reduce the predicted accuracy of the inversion they are clearly-stated and well-understood. The resulting model appears to be fit-for-purpose in identifying potential former pathways and obstacles for ice flow – specific questions below.

Line 154: Not clear where 100 m figure comes from – can you offer a more detailed assessment of uncertainty?

Line 187: the conflict between gravity-predicted and altimetry-observed grounding lines in the region described casts some doubt on the absolute depth values from the gravity inversion. While I agree that the shape of the bed has probably been properly described from the gravity, it would be useful to see on the map what areas were constrained by ship and what weren't, and whether the magnetic data identify the ridge structures as geological structures or whether these reflect surface morphology only. Is there evidence from the acoustic mapping that the gravity features do have bathymetric expression?

Line 241: What is the evidence for ice shelves occupying the Brunt Basin following grounding line retreat? Is it just that they exist in the present or are you referring to geological records?

Technical corrections: Line 34: Be specific that it is the ice shelf front that is retreating.

Line 65: include reference to Figure 1 as well as 2c

Line 82: "has lead to further changes in velocity" or "preceded further changes in velocity"?

Line 106: "includes" – to keep tenses straight

Line 182: Ice Stream (not Steam)

Line 106: be clear where radar data are available (cf line 167 "as three are no direct measurements of ice thickness")

Line 194: the west face of the Bank looks steeper in the figure – is this a trick of shading?

Line 200: I would help to show ice velocity on one of the figures.

Line 227: May *be* the product of glacial planation

Line 272: There is no further increase in thickness that can be attributed to firn or marine ice, but does the thickness still increase?

Line 279: What flow line analysis are you referring to?

Line 336: "upstream" rather than "east" is more consistent

---

## Author Comment (AC1) · 5 Dec 2018

**Response to reviewers**

Reviewer comments: *italics*

Response: plain font

**Reviewer 1**

*i) A few aspects of the methods need clarification.*

*Stage 4 of the bathymetry reconstruction suggests the model is constrained by observational data; how do these data differ from what was used as input in Stage 1? What is left to tighten the inversion?*

The data used in stage 1 is a grid of bathymetry interpolated between the known bathymetric data points. In contrast, only the point data is used as a bathymetric input in stage 4. After steps 1-3 the gravity derived bathymetry should be a good match to the general shape and amplitude of the bathymetry. However, shifts/errors due to un-modelled geology may be present which incorrectly move the inversion away from the true values. Step 4 ensures the bathymetric model accurately matches the constraining observations where they are available, hence giving the best possible integrated model.  Note if the gravity model was perfect there would be no residual shift at stage 4.

Revised text:

> Estimation of the sub-ice shelf bathymetry from the gravity data used a four-step procedure (for details see Supplementary material, Section 1). First, to initiate the inversion we used interpolated grids of ice surface, sub-surface topography and bathymetry from direct observations (Fig. 3 and 4a). The gravity effect of these surfaces was calculated and subtracted from the compiled free-air gravity anomaly (Fig. 4b) to give an estimated Bouguer anomaly. The second stage was to isolate the gravity signatures in the Bouguer anomaly due to bathymetry not described by direct observation. A low pass (150 km) filter isolated signatures due to crustal thickness variations. A ~50 mGal anomaly north of the location of Halley VIa Research station (Fig. 4b) is interpreted from magnetic and gravity data to be a large mafic intrusion 80 km long, 30 km wide and ~6km thick (Jordan and Becker 2018). The gravity anomaly from this structure significantly distorts any bathymetric inversion. A 3D gravity model of this body was therefore constructed. Both the long wavelength crustal anomaly and the gravity model of the mafic intrusion were subtracted from the Bouguer anomaly. This provides a residual gravity anomaly due only to un-modelled variations in sub-ice shelf bathymetry, and un-modelled upper crustal geology. The third stage of the inversion process converts the residual gravity anomaly to variations in bathymetry using the Bouguer slab formula. Adding the calculated bathymetric variations back to the initial bathymetric grid gives a preliminary gravity-derived estimate of bathymetry. The fourth and final stage of the inversion process forces the inverted bathymetry to match observational point data where it is available, providing the best possible bathymetric model (Fig. 3). The final bathymetric model (Fig. 4c) retains uncertainties due to un-modelled geological structures; however, it reveals the broad structure of the sub-ice shelf bathymetry and subglacial topography.

*Can you clarify how the ice shelf base/draft was determined? Line 106 and line 210 both suggest radar data were used; line 166 (Methods) suggests only that shelf thickness (draft) was determined using an assumed density profile and the freeboard height.*

*Are density assumptions valid, since the shelf is partly composed of (fused by) marine ice?*

We have revised the text as follows:

> Although airborne radar measurements of ice shelf draft were available, it was difficult to accurately determine ice shelf draft with these methods. Instead, the ice shelf draft along selected flow lines was calculated based on assumptions of a floating ice column and the density profile of the ice, to translate the freeboard into ice thickness. A common mid-point survey (CMP) of the top 30 m constrained the mean density in the firn pack to 750 kg/m3, and nominal ice densities of 920 kg/m3 were assumed below 50 m. These densities were applied uniformly across the ice shelf without local corrections for incorporated icebergs.

*Explain how Fig 5 shows that contact with the MIR is limited to <1.3-3 km2 (line 215).*

This estimate is based on the area of deformation. We have revised the text as follows:

> The DEM analysis of the MIR shows that the area of deformation caused by contact between the base of the ice shelf and the McDonald Bank is limited to an area of less than 1.3 to 3 km2 (Fig. 5). Strain rates upstream of the MIR (Figure 8 in De Rydt et al., 2018) show the compressive regime in the ice shelf resulting from grounding in 2015.

*Fig 5 appears to show anomalies in surface elevation between different datasets (lidar flight lines vs stereo-image DEM), particularly closer to the grounding line – can you comment on or compare the (un)reliability of these datasets? If the ice surface heights are data-dependent, what does this mean for draft calculations?*

The LiDAR is very reliable. The largest error is an offsets due to tidal effects, which we have not corrected for. This will be a few meters at most. The DEM was a combination of Worldview2 and Cryosat2. We have now replaced the Cryosat 2 coverage (where the biggest discrepancy with the LiDAR data was found), with the latest REMA topography. As can be seen from the image, the match is much better especially near the grounding line, to the extent where you can't actually see the LiDAR lines very well.

*Which velocity measurements or model were used to convert distance to time in Fig 6? Could simply be stated in the figure caption.*

This is now stated in the Figure caption as suggested.

*ii) Former grounded ice reconstruction*

*It would be useful to see the seismic profile across the McDonald Bank (line 61, line 225 – is it possible/permitted to reproduce this figure?) and also where, specifically, it crosses the bank (e.g. to inform your comment on line 228-229). What direction do these bedded glacial sediments dip in? What grounded ice geometry would be consistent with the seismic structure – your interpreted grounded ice flow trajectory (Fig 4) would suggest the bank was lateral to ice flow reaching the*

*continental shelf edge, but is now perpendicular to flow from the shelf (switch from lateral to ice frontal position).*

*I wonder whether there are sufficient seismic data (coverage or resolution?) to detect this switch, or to evaluate your reconstruction? Are there any structures related to the re-grounding of the Brunt Ice Shelf on this bank? Can different generations of sedimentation be detected?*

The seismic data were acquired by the Norwegian Antarctic Research Expedition in 1976/77 and 1978/79. The original data are no longer available, but instead are presented as crude line drawings based on sparker data. To better incorporate this data we revise the text as follows:

> At these times, glacial depositional processes operating between the ice streams occupying Filchner Trough and the Brunt Basin likely formed the layered glacial sediments of the McDonald bank interpreted from seismic surveys (Elverhøi and Maisey, 1983). Although seismic lines at different orientations are required to fully characterise the internal architecture and processes forming this deposit, the line drawings based on sparker data presented by Elverhøi and Maisey (1983, Profile 5, page 486) suggest the layered glacial sediments dip westwards into the Weddell Sea.  At least two of these layers (Units 1 and 2) were interpreted as being of 'glacial origin', but whether they are 'glaciomarine sediments, till and/or glacially compacted glaciomarine sediments, cannot be determined'.  These layers have subsequently been truncated along the steep eastern slopes of the McDonald Bank, presumably by erosive ice advances along the south-to-north oriented trough under the BIS. The relatively flat top of the McDonald Bank may the product of glacial planation processes by ice shelves during, and either side of peak glacial conditions.

*Line 185-90: I suggest removing the references to grounding line positions from the series of brackets here, and making your interpretations separately. i.e. report in these few sentences where the topographic highs are, and then finish the paragraph stating that you interpret these high points as likely pinning points, perhaps both for grounded ice retreating from its maximum extent (grounding line positions) and stabilisation points for subsequently floating ice. On first reading it seemed as if you were calling on independent evidence of grounding line positions that you now match up with your new bathymetry, but I don't think that's the case.*

We have followed this suggestion and rewritten this section as follows:

> A number of topographic highs occur in the Brunt Basin. Two of these are high enough to make contact with the base of the SWGT at the Lyddan Ice Rise, and the base of the BIS at the McDonald Bank. Other topographic highs are present in the NNW oriented part of the Brunt Basin under the SWGT including a series of at least three transverse ridges on the flanks of the trough (marked as inferred grounding lines 1-3 in Fig. 4c). Our gravity-derived topography predicts these ridges are in contact with the base of the SWGT, which is not indicated by ice velocity or satellite imagery. Instead, we suggest that these ridges fall just short of the base of the ice. Two similar transverse ridges at the present day grounding line in the East-West oriented part of the trough beneath the SWGT (marked as inferred grounding line 4 in Fig. 4c). We interpret these topographic highs as likely pinning points, perhaps both for grounded ice retreating from its maximum extent (grounding line positions) and stabilisation points for subsequently floating ice.

*Section 4.1: since the only evidence of glacial advance/retreat that could be considered diagnostic in this new dataset are the convergent troughs, the tone of this section is somewhat over-confident and some of the assertions cannot (yet) be supported. These assertions may well be likely but specific*

*landform or sedimentological evidence is still absent, so I would rephrase the language more towards presenting likely hypotheses than definitive conclusions.*

We have changed the first sentence to state:

> The bathymetry and subglacial topography provide sufficient evidence to propose a range of past ice configurations in the study area.

Elsewhere we have used more careful wording – separating inferences from evidence of past ice configurations. Please see the extensive tracked changes in the revised text.

*iii) Discussion – does not evaluate the future development of the Brunt Ice Shelf, as the article introduction states as its aim (line 94), only outlines four possibilities. A paragraph or so of synthesis seems to be lacking here, and I think there is some room for evaluation without going too far out on a limb. Whichever of the four scenarios will develop depends, at least in part, on feedbacks between ice shelf grounding on a distal pinning point, the ice shelf flow regime, the production of ice shelf (berg) keels, the draft and the packing of those keels (inherited from the shelf-sheet grounding line?) Such feedbacks are alluded to through the manuscript, but are not drawn together to inform some of the possibilities that the Discussion scenarios raise.*

To address this suggestion we have added the following paragraph:

> Although the outcome of the calving of the ice shelf is not yet known, these four scenarios all show the importance of understanding the geometry the ice shelf and the bed. Which of the four scenarios will develop depends, at least in part, on whether the ice shelf is able to remain in contact with one of the topographic highs on the McDonald Bank. Whilst the depth of the topographic highs is fixed, the contact between the ice shelf and the bed depends on a number of variables. These include the short-term development of the ice shelf flow regime following calving, and its impact on ice shelf draft (e.g. lateral spreading) and structural integrity (e.g. development of further rifts resulting from changes in the strain rate). They also include the long-term influences of processes at the grounding line, such as the thickness and velocity of ice flowing across the Brunt Icefalls which determine the spacing and depth of the iceberg keels.

*Line 325-9: the SWG is fed by a much larger (and faster flowing?) supply basin, so presumably sustains a significant ice tongue partly due to high input, even though it's unbuttressed. Does the very different supply catchment for the BIS not make this a poor analogue?*

The velocity data added to Fig. 5 suggest rather similar flow rates for the BIS and SWGT

*How limited do you think your conclusions or interpretations of future ice shelf development are by the specific three flowlines you have chosen to analyse shelf draft? This is the broad flowline reaching the shallowest part of the McDonald Bank; is it also the deepest draft zone of the shelf? Could keels along other flowlines eventually reground on other topo highs on the McDonald Bank?*

Yes – we have now included this in scenario 3.

*Figures, annotation & captions*

*Figure 1. I wonder about the choice of satellite imagery – these show the geography clearly, but don't visualise the structures as well as, for example, Fig 1 in King et al 2018.*

This is more recent satellite imagery than King et al 2018. We have used blue shading to highlight the key structures (Chasms 1 and 2 and Halloween Crack).

*Figure 3: several missing features either in annotation or caption - I don't see any shading corresponding to swath bathymetry or other sonar data (pale blue shading, in the caption).*

*- the white lines – calving line and grounding line (should be stated in caption) – from which source and year?*

*- what are the grey blobs offshore? Iceberg outlines? From what source/year, and are they really needed (could mask out)?*

*- could you separate the seismic sites for water depth estimates from the historical depth soundings (use different symbols)?*

*- what's the pink triangle? Halley VIa?*

We have re-formatted the figure and caption to address these points. Firstly, we now highlight the areas with swath data coverage more clearly in red. The grey blobs were simply areas with no swath coverage – not icebergs. The seismic and historical depth soundings are now separated. The coastline/grounding line is from the 2013 Antarctic Digital Database, now noted in the caption.

Revised caption for Figure 3:

> Figure 3. Data locations overlain on a MOA satellite image. Yellow lines indicate 2017 aerogeophysical flights collecting radar, gravity and magnetic data. Red shading indicates swath bathymetry data coverage. Blue lines mark radar depth determinations from BEDMAP2 (Fretwell et al., 2013). Orange lines mark location of ICEGRAV 2013 radar and gravity data (Forsberg et al., 2017). Black dots mark seismic determinations of water column thickness under the Brunt ice Shelf (Hodgson et al., 2018). White dots mark soundings from historical ship tracks acquired when the ice front of the Stancomb-Wills Glacier Tongue was less-advanced. White line marks the grounding line/ice shelf edge from the 2013 Antarctic Digital Database. Pink triangle marks approximate location of Halley VIa station from 2012 to 2016.

*Figure 4: missing or unclear features in annotation or caption - the depth scale isn't particularly straightforward to match to the shading on the panels, which appears more intense and has hillshade effects. Try a classified colour scale, rather than ramped, perhaps? Or make it clearer at what depth the colour bar saturates and/or at what depth the shelf break is, for reference. Also, consider shifting the bluebrown shift to 0m, rather than the unintuitive brown = submarine as well as terrestrial.*

*- add a label for 4d to the box in 4c*

*- note the black dots = seismic soundings*

*- remove the iceberg(?) polygon outlines, these don't seem to be necessary here*

*- what are the green outlined features?*

*- label Fig 6 flowlines on the panel itself (rather than in caption)?*

*- flow arrows and grounding lines are both hypothesised. Suggest something like 'Probable orientations of fast grounded ice flow: : :' and 'Inferred former grounding line positions based on existence of topographic highs: : :'*

*4d/5: use either decimal degrees or degrees & minutes (latitude labels) consistently on all figures.*

*5: label Chasm 1, 2, Halloween Crack to better follow discussion.*

*6: suggest label/arrow ice flow direction.*

We have remade the topographic parts of Figure 4 using a revised colour scale. This scale saturates at +/-1000m and has a central green/blue transition at zero meters elevation. The box and profiles are now ladled with the appropriate figure numbers. We now note the black dots are seismic stations, and the thin black lines locate the edges of the swath bathymetric data coverage (not icebergs). We have included the suggestions for describing the arrows and former grounding lines in the figure captions, and now note that the green lines circles and lines indicate possible past and present pinning points. Figures 3 and 4 now only use whole degrees.

Revised caption for Figure 4:

> Figure 4. Revised topography beneath the Brunt Ice Shelf and Stancomb-Wills Glacier Tongue. (a) Topography derived from direct observations including; swath bathymetry offshore and in areas historically accessible to ships during past calving events, seismic depth sounding of the ice shelf, and radar depth sounding over the grounded ice sheet. (b) Free air gravity anomalies. Data inside black outline from new strapdown gravity data set (Becker et al., 2018). Regional data from ICEGRAV 2013 survey (Forsberg et al., 2017) and previous regional compilations (Jordan et al., 2017). Note gravity 'high' outlined in yellow attributed to a large mafic intrusion based on gravity and magnetic signatures (Jordan and Becker, 2018). (c) Integrated bathymetric model including additional constraints from gravity data beneath the ice shelf. Black and white contour is the predicted grounding line. Thin blue contours show areas predicted to be grounded by ice 50, 100 or 200 m deeper than the calculated bed. Black contours show 50, 100, 200m predicted cavity thickness. Probable orientations of past grounded ice flow are indicated by white arrows. Inferred former grounding lines based on mapped topographic highs are marked by numbered black dashed lines (1-3). (d) Detail of Brunt ice shelf. Red lines mark the position of flow-lines plotted in Fig 6. Black dots show seismic stations and thin black lines mark edges of swath bathymetric data coverage. The pink triangle marks the position of Halley VIa Research Station. Green circles and lines indicate possible past and present pinning points. Abbreviations: LIR (Lyddan Ice Rise), SWS (Southern Weddell Sea), MB (McDonald Bank), SWGT (Stancomb-Wills Glacier Tongue), BIS (Brunt Ice Shelf), BB (Brunt Basin), DLGT (Dawson-Lambton Glacier Tongue).

---

## Author Comment (AC2) · 5 Dec 2018

**Response to reviewers**

Reviewer comments: *italics*

Response: plain font

**Reviewer 2**

*Scientific questions: Line 61: I assume this bank is hard-rock cored. Was the sediment thickness from seismic surveys used in the gravity bathymetry model?*

It is unclear from the sparse relatively old seismic data if this bank is hard-rock cored or not. No sediment thicknesses from seismic data were used in the gravity inversion for bathymetry as this information was not available. However, magnetic depth to source data, and a coupled significant positive gravity anomaly in this region indicate a significant sub-surface dense body. The impact of this body is removed, based on a relatively simple 3D gravity model, prior to carrying out the bathymetric inversion. We note that the relatively large number of seismic tie points and offshore swath bathymetric data constrain the bathymetric model in the Brunt region. Discrepancies in the bathymetric results due to un-known sedimentary basins etc. should therefore be relatively limited.

*Line 133 to 157: The gravity inversion technique presented here is novel, and while it makes assumptions that will reduce the predicted accuracy of the inversion they are clearly-stated and well-understood. The resulting model appears to be fit-for-purpose in identifying potential former pathways and obstacles for ice flow*

*Line 154: Not clear where 100 m figure comes from – can you offer a more detailed assessment of uncertainty?*

It is challenging to provide a full assessment of uncertainty, as step 4 of the inversion ensures the bathymetric model matches any independent observations perfectly. The 100m error estimate stems from how much the assumption of a floating ice shelf is breached, which we consider a reasonable minimum estimate for the amount of error. Following the reviewers suggestion we provide a more detailed discussion of the possible errors and suggest an error distribution with a standard deviation of ~175 m represents a reasonable maximum error estimate.

Revised text L151-157

> Uncertainties arising from unknown and un-modelled geology are hard to quantify, as step 4 of the inversion means the model always matches the direct bathymetric observations. One estimate of the errors due to geological factors can be made by looking at the difference between the initial gravity inversion (step 3) and the direct observations. This reveals a symmetrical error distribution with ~0 mean, and a standard deviation of ~175 m, which we attributed to un-accounted geological biases, and the long wavelength of the regional gravity data. This therefore represents a worst case estimate of the expected error far from control points. Step 4 will have in-part accounted for the impact of unknown geological features, and hence reduced the overall error of the resulting inverted bathymetry. One alternative check is to compare predictions of ice shelf flotation, based on freeboard and an assumed ice shelf density, to the final inverted bathymetry (Fig. 4c). This reveals that the inversion results generally predict the grounding line well. A key discrepancy is beneath the

SWGT at 75°S, where flotation is violated by 50-100 m. We therefore consider +/- 100 m a reasonable minimum estimate of the error in the bathymetric inversion in this region. Northeast of the 2017 survey area, the inversion suggests a broad area of ice shelf should not be floating. We attribute this to a lack of high quality data coverage, and/or actual observations of bathymetry.

*Line 187: the conflict between gravity-predicted and altimetry-observed grounding lines in the region described casts some doubt on the absolute depth values from the gravity inversion. While I agree that the shape of the bed has probably been properly described from the gravity, it would be useful to see on the map what areas were constrained by ship and what weren't, and whether the magnetic data identify the ridge structures as geological structures or whether these reflect surface morphology only. Is there evidence from the acoustic mapping that the gravity features do have bathymetric expression?*

The extent of the swath bathymetry data is now better highlighted in Fig. 3, and is also shown in Fig. 4d. The onshore topography is also relatively well constrained by airborne radar measurements. Comparison of bathymetry/topography from direct observations only (Fig. 4a) and the free air gravity anomaly (Fig. 4b) highlights the fact that major structures such as the deep onshore basin, and the deep trough beneath the BIS are both resolved by these independent data sets. In other areas there are no direct measurements coincident with the high resolution gravity data to provide further checks on our results.

We have added an image of the aeromagnetic data across the survey region to the supplementary material. The first thing that we note from this data is that there is a clear high frequency signal beneath the main survey area. This suggests the geological basement is close to the surface, and a major thick sedimentary basin which could distort the results is not present. In addition, no clear 1:1 correlation between the inverted bathymetry and the underlying geologic (magnetic) fabric is seen. This suggests the bathymetry dominates any underlying geological signal. However, the northern part of topographic ridge 3 does appear to follow a relatively short wavelength negative magnetic anomaly, indicating geological control of this structure. The more limited data coverage in this region (Fig. 3) makes further detailed discussion of the underlying geological origin of this feature problematic.

We have added the following additional paragraph after L157 to address this point:

> In regions where both direct topographic/bathymetric observations and high resolution gravity are available (Fig. 4a and b respectively) major topographic structures, including the deep onshore basin and the trough beneath the BIS, are resolved by the gravity data. This supports the use of gravity data to fill the intervening areas where no direct measurements are available. Aeromagnetic data across the study area (SFig. XX) shows a clear high frequency signal beneath the main survey area. This suggests the geological basement is close to the surface, and a major thick sedimentary basin which could distort the results is not present. In addition no clear 1:1 correlation between the inverted bathymetry and the underlying geologic (magnetic) fabric is seen. This supports the view that the bathymetry dominates any underlying geological signal. However, the northern part of topographic ridge 3 does appear to follow a relatively short wavelength negative magnetic anomaly, indicating geological control of this structure. The more limited data coverage in this region (Fig. 3)

makes further detailed discussion of the underlying geological origin of this feature problematic.

*Line 241: What is the evidence for ice shelves occupying the Brunt Basin following grounding line retreat? Is it just that they exist in the present or are you referring to geological records?*

We have changed the sentence as follows:

Following grounding line retreat, floating ice likely occupied the Brunt Basin, as it does today.

*Line 194: the west face of the Bank looks steeper in the figure – is this a trick of shading?*

We now also refer to Fig. 6 where the profile is presented.

*Line 200: I would help to show ice velocity on one of the figures.*

This data is now added to Figure 5.

All other minor comments addressed

---

## Author Response (AR2)

To the editorial team

We thank the editor for their decision and additional suggestions for minor revisions. We have taken on-board and carried out all the suggested changes. A full point by point response is included below – our response as red in track changes.

===

Editor Decision: Publish subject to minor revisions (review by editor) (20 Dec 2018) by Joseph MacGregor

Comments to the Author:

Dear Dr. Hodgson et al.,

Thanks for your revised manuscript and response to the two reviewers' comments. Your response is thorough and thoughtful, further strengthening your submission. This submission is timely, given the Brunt Ice Shelf's apparent vulnerability, and you provide the context needed to better understand its future. I learned quite a lot reading it, and the expert reviewers are also generally satisfied with it.

Below I have several minor comments for your consideration prior to a final decision on your manuscript. My main concern at this stage are the quality of the figures, which in general could be improved, but I have a couple of broader questions also.

Thanks again for considering The Cryosphere for your work,

Joe MacGregor

NASA GSFC

103: Here are elsewhere, it is mentioned that the future of the BIS depends on the propagation direction of Chasm 1, but it is never stated what direction it is currently propagating and therefore which scenario might carry greater weight. This is a missed opportunity. Of course, any mention thereof should be carefully qualified, because that direction could change, but as it stands it introduces an unnecessary degree of ambiguity into the manuscript.

The direction of crack propagation towards the centre of the MIR (as predicted by De Rydt 2018) is now stated at this point in the text and the manuscript has been updated for clarity.

151-3: I am confused by the description of this final step of the inversion here and in the supplementary material. To solve inverse problems, one generally does not seek a model that fits the data (where available) exactly, as that risks over-fitting the data. It's more prudent to seek the minimum amplitude correction necessary to fit the data within their uncertainty bounds, with a suitable tradeoff factor based on expert assessment of the datasets and forward model. This concern seems particularly relevant given the different spatial scales that aerogravity data are

sensitive to as compared to seismic and swath bathymetry. Please assess this concern or at least state clearly in the text whether the uncertainty in the data is effectively ignored.

We recognise that the final step of our procedure to estimate the bathymetry across the study area does not match that for the typical formal solution of an inverse problem. However, our aim is to produce a bathymetric model which both matches the direct measurements of bathymetry and is guided by the gravity data between. As noted, the discrepancy in wavelengths resolved by the direct topographic observations and gravity datasets complicates this, and any result solely from a formal inversion of gravity would not precisely match the topographic tie points. The final stage in our estimation process should therefore be considered as a method of blending two distinct measurements/estimates of bathymetry, rather than an inversion. The text and supplementary material have been updated to clarify that what we have developed is a procedure (algorithm) for estimating bathymetry, rather than a formal inversion.

Figures:

In general, I find figures are better served long-term if they can be interpreted as independently as possible from their captions. This requires both legends and labels. Presently, the authors favor a caption-heavy approach, but I encourage you to explicitly reconsider whether that is the most appropriate approach for your figures in the long-term.

We recognise that there are different stylistic preferences for the amount of text to be added to figures. As we feel that excessive labelling can clutter the figures we have added keys to Figures 3 and 4 to highlight the most significant points and aid comprehension independent of the caption. Other figures remain un-changed (aside from in response to the points below).

Please use a consistent symbol for each station throughout the figures.

A consistent symbol and location (red square) is now used for Halley VIa.

Figure 1: Label panels (a) and (b). Is the light blue where the rifts are entirely artificial? I'm assuming it is, as I do not recall such unusual coloration in my recent flyover of this region, but it's not mentioned in the caption. Separately, despite the graticule, an inset map showing the location in Antarctica would be appropriate to include here.

Panels are now labelled, the colour shading of the rifts is noted (as noted this is entirely artificial), and an inset map showing the study area in its Antarctic context has been added.

Figure 2: These photos really need labels and their approximate locations ought to be labeled in Figure 1a.

Photographs have been labelled and the approximate locations of 2a-c added to Figure 1. Photograph 2d was taken to the south of the study area, - also now noted in the caption.

Figure 4. The color range used for the bathymetry undersells the range of topographic features that is discussed in the manuscript, especially the inferred former grounding lines. Further, the thin blue and black lines representing aspect of the sub-ice-shelf cavity and bathymetry are nearly invisible. Please reconsider colors used in this figure.

The colour-scale has been expanded a little in the offshore areas to reveal more detail. In addition the contours showing the thickness of the cavity relative to an isostatic model of the floating ice shelf have been re-coloured, and a key has been added.

Figure 5. I like the way the comparison between the airborne and satellite data is done. Label physical property represented by color bar. Label meaning of flowlines (that they're shown in Figure 6). Because of the physical importance of the small area where the BIS still contacts the McDonald Bank (and the weight that the manuscript lends to it), an inset panel should be generated that zooms in on that region. As it stands, I think they're only shown (but without being referred to in the text as such) in Figure 4?

Physical property of colour bar (surface height above geoid) now labelled. Flow lines now labelled as Fig. 6 in key and noted in caption. Inset showing zoom of MIR is now added, including full swath Lidar surface elevation grid (caption updated to match).

Figure 6. Instead of solid/dotted/dashed lines, for which the latter two are nearly indistinguishable, used three shades of the same color to distinguish between the different flowlines. "Distance from MIR" should instead by "Distance upstream of MIR". In panel a, label ice flow direction.

Lines now separated by colour, "distance upstream" added, and ice flow direction included.

Text:

29: ice shelf depends on OK

38: 74ºS OK

40: 80 km OK

97: including: OK

102: outcomes depend initially on OK

125: is ground, airborne OK

126: aerogeophysical OK

132: no need to qualify data as "innovative" if they have already been described in a previous study OK

230-1: clarify that the "orientation" of the icebergs presumably refers to their long axis? OK

232: was examined in detail along OK

257: McDonald Bank OK

289: is neither thick nor extensive OK

326-7: values of depths should not be negative OK

341: of the BIS depends not only OK

408: …in a region of Antarctica that is not presently experiencing a rapid increase in atmospheric temperature or a known intrusion of circumpolar deep water, the BIS… OK

408-417: merge paragraphs OK

416: Chasm 1 OK

Typos in Supplementary Material:

- Cochran not Cochrane ok

- kg m–3 not kgm–3 ok

- respectively not respectivley ok

[revised manuscript text omitted]

**Step 1. Initiate.** The 3D gravity effect of the initial known topography, interpolated from radar, seismic and swath bathymetry measurements, is calculated. This Bouguer correction is subtracted from the observed grid of free air gravity anomalies. The resulting Bouguer anomaly contains signals caused by variations in crustal thickness, shallow geological bodies with distinct densities, and errors in the input topographic data.

**Step 2. Isolate.** The long wavelength gravity anomaly, assumed to be due to crustal thickness variation, is isolated using a low pass filter. This long wavelength anomaly is subtracted from the Bouguer anomaly, leaving the 1st residual gravity anomaly, which contains signatures due to shallow geological bodies with distinct densities, and errors in the input topographic data. If distinct geological features can be recognised 3D models of their gravity signatures are calculated and subtracted leaving the 2nd residual gravity anomaly, theoretically only due to errors in the input topographic data.

**Step 3. Calculate.** The 2nd residual gravity anomaly is converted from mGal to equivalent rock thickness using the Bouguer slab formula, assuming a density of 2.67 and 1.028 gcm$^{-3}$ for rock and water respectively. This

equivalent topographic correction is added back to the initial topography to provide the 1st estimate of gravity improved topography.

**Step 4. Constrain.** Discrepancies between the 1st estimate of sub-ice shelf topography and the direct point observations of topography relate to un-modelled geological factors, and features resolved by higher resolution topographic datasets. We interpolate this error field between the points where topography is well constrained to provide a residual topographic correction across the study area. This is added to the 1st estimate of gravity improved topography to provide the final blended gravity improved topography estimate.

[Figure]

**Supplementary Figure 1. Flow diagram showing recovery  of bathymetry from gravity. Right-hand panel shows example grids at each stage (red-green-pink colours = gravity (mGal), blue-brown colours = topography (m)). Note Pink triangle marks original Halley VI site.**

Specific notes for this study:

The Bouguer correction was calculated using a 3D Gauss-Legendre quadrature (GLQ) method (von Frese et al., 1981), assuming a uniform observation altitude of 450 m, coincident with the 2017 survey altitude over the ice shelf. Results are therefore not valid onshore where surface elevations are above 450 m. Standard densities for the Bouguer correction of 915, 1028 and 2670 kg m$^{-3}$ for ice, water and rock respectively  were used.

The residual crustal anomaly was isolated using a 150 km low pass Gaussian filter. This appeared to remove any linear regional trend in the observed gravity anomaly, suggesting the long wavelength signal due to the extreme change in crustal thickness between the East Antarctic continent and the oceanic crust of the Weddell Sea have been accounted for.

For areas outside the high-resolution 2017 data anomalies appear in the Bouguer gravity field which are likely artefacts of the low resolution of the reconnaissance free air gravity data.

The underlying gravity data set is described and analysed in more detail in (Jordan and Becker, 2018) and the associated gravity data is published through the UK polar data centre (Becker et al., 2018).

**2. Comparison of gravity improved topographic estimate and aeromagnetic data**

Aeromagnetic data across the study area shows a clear high frequency signal beneath the main survey area. This suggests the geological basement is close to the surface (Supplementary Fig. 2).

[Figure]

[Figure]

**Supplementary Figure 2. Comparison of topography and aeromagnetic data. a) Final estimate of gravity improved topography. Note black and white dashed lines (1-3) marking potential former grounding lines which likely acted as pinning points. Solid black outline marks edge of 2017 gravity survey where gravity derived topography is most robust. Red square marks Halley VIa site. b) Aeromagnetic data (Jordan and Becker, 2018; Jordan et al., 2018) over the study area. A direct regional correlation between topography and the underlying geological structures revealed by the magnetic data cannot be made, indicating the bathymetric signal likely dominates the geological signal. However, local correlations can be seen, for example between the northern part of ridge 3 and a linear magnetic low. Such correlation suggests that subglacial geology is influencing the bathymetry, but relatively sparse data coverage makes more detailed analysis difficult. Note yellow outline marks gravity high (Fig. 4b in main text) associated with very high amplitude magnetic anomalies (>1000 nT) attributed to a mafic intrusion (Jordan and Becker, 2018).**